# Multiple selection filters ensure accurate tail-anchored membrane protein targeting

Meera Rao[1], Voytek Okreglak[2,3†], Un Seng Chio[1†], Hyunju Cho[1], Peter Walter[2,3], Shu-ou Shan[1*]

[1]Division of Chemistry and Chemical Engineering, California Institute of Technology, Pasadena, United States; [2]Howard Hughes Medical Institute, University of California, San Francisco, United States; [3]Department of Biochemistry and Biophysics, University of California, San Francisco, United States

**Abstract** Accurate protein localization is crucial to generate and maintain organization in all cells. Achieving accuracy is challenging, as the molecular signals that dictate a protein's cellular destination are often promiscuous. A salient example is the targeting of an essential class of tail-anchored (TA) proteins, whose sole defining feature is a transmembrane domain near their C-terminus. Here we show that the Guided Entry of Tail-anchored protein (GET) pathway selects TA proteins destined to the endoplasmic reticulum (ER) utilizing distinct molecular steps, including differential binding by the co-chaperone Sgt2 and kinetic proofreading after ATP hydrolysis by the targeting factor Get3. Further, the different steps select for distinct physicochemical features of the TA substrate. The use of multiple selection filters may be general to protein biogenesis pathways that must distinguish correct and incorrect substrates based on minor differences.

*For correspondence: sshan@caltech.edu

†These authors contributed equally to this work

Competing interests: The authors declare that no competing interests exist.

## Introduction

Efficient and accurate protein localization is a prerequisite to generate and maintain compartmentalization in all cells. Understanding how protein-targeting pathways achieve accurate membrane protein localization has been challenging for multiple reasons. First, topogenic signals that define a protein's final destination tend to be degenerate and lack consensus motifs (*von Heijne, 1985*; *Zheng and Gierasch, 1996*), which demands targeting machineries to be adaptable and able to recognize a diverse set of signals. Second, minor differences in targeting signals distinguish proteins that belong to alternative pathways or organelles (*Emanuelsson and von Heijne, 2001*; *von Heijne, 1985*; *Zhang and Shan, 2014*; *Zheng and Gierasch, 1996*). Thus, protein-targeting pathways must also evolve robust selection mechanisms that can detect these minor differences. Furthermore, hydrophobic transmembrane domains (TMDs) on membrane protein substrates are prone to irreversible aggregation that can lead to mislocalization and proteostatic stress, requiring targeting machineries to also effectively shield the TMDs during targeting (*Shao and Hegde, 2011*). Except for a few systems (see [*Randall and Hardy, 1995*; *Zhang and Shan, 2014*]), the molecular mechanisms by which protein targeting machineries overcome these challenges are not well understood for most pathways.

A salient example of these challenges is an essential class of tail-anchored (TA) membrane proteins, defined solely by a single TMD near the C-terminus. TA proteins comprise 3–5% of the eukaryotic membrane proteome and play essential roles in numerous processes including membrane fusion/fission, vesicular trafficking, protein translocation, quality control, and apoptosis (*Beilharz et al., 2003*; *Hegde and Keenan, 2011*; *Kalbfleisch et al., 2007*). TA proteins are found in nearly all membranes in eukaryotic cells including the endoplasmic reticulum (ER), the mitochondrial outer membrane (OMM), and peroxisomes (*Kutay et al., 1993*; *Yabal et al., 2003*). To a first

approximation, sequences near the C-terminus of TA proteins (including the TMD) are necessary and sufficient to direct their proper localization to diverse organelles (*Beilharz et al., 2003*; *Whitley et al., 1996*). Previous work further showed that modulation of the TMD and basic residues at the extreme C-termini alter the localization of TA proteins to the ER or mitochondria (*Beilharz et al., 2003*; *Borgese et al., 2007*, *2003*; *Rapaport, 2003*). Nevertheless, the molecular mechanism by which these sequences are recognized and decoded by protein targeting machineries remains poorly defined.

Recent advances in deciphering TA protein targeting pathways provide an opportunity to address this question. Biochemical and genetic analyses have identified the Guided Entry of Tail-anchored protein (GET) pathway, which targets TA proteins destined to the ER through a series of substrate handoff events (*Schuldiner et al., 2008*; *Stefanovic and Hegde, 2007*; *Wang et al., 2010*). The co-chaperone Sgt2 initially associates with the TMDs of TA proteins (*Wang et al., 2010*). A scaffolding complex, comprised of Get4 and Get5 (Get4/5), bridges Sgt2 and the central targeting factor, the Get3 ATPase (*Chartron et al., 2010*, *2012*; *Wang et al., 2010*). Get4/5 also preorganizes Get3 into the optimal conformation and nucleotide state for TA binding (*Gristick et al., 2014*; *Rome et al., 2013*) and thus facilitates the transfer of TA substrates from Sgt2 to Get3 (*Wang et al., 2010*). TA loading on Get3 drives Get3's dissociation from Get4/5 and also activates Get3's ATP hydrolysis activity (*Rome et al., 2014*, *2013*). After ATP hydrolysis, the Get3•TA complex associates with the Get1/2 receptor complex on the ER membrane, via which the TA protein is released from Get3 and inserted into the membrane (*Schuldiner et al., 2008*; *Wang et al., 2014*).

It has been established that a highly hydrophobic C-terminal TMD promotes the association of a TA protein with Sgt2 (*Wang et al., 2010*) or the mammalian Get3 homolog TRC40 (*Mariappan et al., 2010*). Nevertheless, it is unclear how the GET pathway distinguishes TAs destined to different organelles and selects the correct set of substrates; it is also unclear how Sgt2, Get3, or the target membrane contributes to this selection. Here, we address these questions by systematically varying the TA protein and quantitatively analyzing how individual molecular steps in the GET pathway sense and respond to these variations through a combination of biochemical, biophysical, and cell biological studies. Our results show that the GET pathway selects ER-destined TA proteins using multiple steps in the pathway, and different steps are capable of recognizing distinct physicochemical features in the TA protein. The combination of these mechanisms together ensures accurate substrate selection by the GET pathway.

## Results

### TAs are targeted to the ER based on features in both the TMD and the C-terminal element

Previous work suggested that highly hydrophobic TMDs direct TA proteins to the ER (*Burri and Lithgow, 2004*; *Kalbfleisch et al., 2007*). A comprehensive analysis of the Grand Average of Hydropathy (GRAVY) (*Kyte and Doolittle, 1982*) scores of TAs shows that: (i) the TMD of TAs span a wide range of hydrophobicity (*Figure 1*, GRAVY Score); (ii) among these, established GET substrates (*Mateja et al., 2015*; *Schuldiner et al., 2008*) (*Figure 1A*, highlighted in black) are enriched in the range of higher hydrophobicity; (iii) mitochondrial TAs tend to span a range of lower hydrophobicity, but exhibit significant overlap with that of GET substrates (*Figure 1B* vs. *1A*). These observations suggest that features in addition to TMD hydrophobicity also dictate the localization of TAs. A potential distinguishing feature is the basic residues C-terminal to the TMD, which has been shown to direct proteins to outer mitochondrial membrane in *Arabidopsis* and mammalian cells (*Marty et al., 2014*; *Borgese et al., 2001*). As natural amino acid variations tend to change multiple physicochemical properties, additional features of the TMD besides hydrophobicity could also contribute to the localization of TA proteins.

To understand how TA substrates are selectively targeted to the ER, we established a set of model TAs in which we independently varied the TMD and the C-terminal element (CTE). As model substrates destined to the ER and mitochondria, we focused on the TMD and CTE of Bos1p (residues 207–244) and Fis1p, respectively (*Figure 2A*). A non-cleavable, N-terminal 3xStrep-SUMO motif was fused to this sequence to enable purification and improve solubility of the substrate (*Figure 2A*; see also [*Wang et al., 2010*, *2011*]). We replaced increasing numbers of hydrophobic

**A**

| Gene | TMD | GRAVY Score | Agadir Score | Localization |
|---|---|---|---|---|
| YDL012c | ASSGNEDCLAGCLAGLCLCC | 0.99 | 0.46 | PM |
| Prm3 | FYQGAIFGSFLGAAVTTVLSNLAV | 1.19 | 0.14 | NE |
| Sss1 | YTKIVKAVGIGFIAVGIIGYAIKLIHIPI | 1.51 | 0.52 | ER |
| YPL206c | WVHIKLCGWSIAYVIFLFL | 1.67 | 0.78 | ER |
| Sec12 | FFTNFILIVLLSYILQFSY | 1.70 | 5.36 | ER |
| Ufe1 | AKMTTYGAIIMGVFILFL | 1.72 | 0.45 | ER |
| Vps64 | VSKGMLFGVVAISFGLVATA | 1.72 | 0.34 | ER |
| Dpm1 | FGANNLILFITFWSILFFYVC | 1.73 | 1.88 | ER |
| Pex15 | VLNKNGLLLTGLLLLLCL | 1.79 | 1.94 | ER/Peroxisome |
| YBL100c | IAAVRANIICACFFYLFCYCS | 1.83 | 0.34 | ER |
| Bet1 | SGISIKTWLIIFFMVGVLFFWVWI | 1.92 | 9.41 | ER-Golgi |
| Frt1 | FIIDIIAFLLMGGFIVYVKNLL | 1.97 | 3.41 | ER |
| Sft1 | SIWRMVGLALLIFFILYTLF | 2.01 | 10.35 | Golgi |
| Kar1 | YFLWTICILILLYCNIYV | 2.01 | 5.43 | NE |
| Sec20 | VYLSLGFLLCCVSWVLW | 2.02 | 1.46 | ER |
| Cyb5 | GSGTLVVILAILMLGVAYYLL | 2.12 | 13.56 | ER |
| Scs2 | SSSMGIFILVALLILVLGWFY | 2.13 | 34.51 | ER |
| Far10 | FTLLTFGTISIGIIAIVFKIL | 2.15 | 0.81 | ER |
| Sec22 | QYAPIVIVAFFFVFLFWWIFL | 2.19 | 0.42 | ER |
| Ysy6 | LGILLFLLVGGGVLQLISYIL | 2.28 | 1.96 | ER |
| Nyv1 | ITLLTFTIILFVSAAFMFFYL | 2.30 | 0.60 | Golgi-Vacuole |
| Csm4 | FVIAELNSLIIVFFISLVFLW | 2.35 | 0.68 | ER |
| Tlg2 | VILLLLTLCVIALFFFVMLKPH | 2.36 | 1.49 | Golgi |
| Gos1 | AFVLATITTLCILFLFFT | 2.38 | 0.98 | Er-Golgi |
| Sed5 | WLAAKVFFIIFVFFVIWVLVN | 2.41 | 1.38 | ER-Golgi |
| Sbh2 | LVVLFLSVGFIFSVIALHLLT | 2.52 | 0.31 | ER |
| Sbh1 | PLVVLFLAVGFIFSVVALHVI | 2.64 | 0.98 | ER |
| Ubc6 | MVYIGIAIFLFLVGLFM | 2.66 | 1.46 | ER |
| Vti1 | FISYAIIAVLILLILLVLFSK | 2.67 | 74.40 | Golgi-Vacuole |
| Phm6 | IIVIIIVLLLYSLTMVGLFYV | 2.79 | 3.97 | Vacuole |
| Frt2 | ALDIVFLIIIIVICYTF | 2.79 | 13.30 | ER |
| Bos1 | LVFWIALILLIIGIYYVL | 2.81 | 13.19 | ER-Golgi |
| Slt1 | LFYITVFIFMILGLVFTFIII | 2.84 | 1.76 | ER |
| Vam3 | CGKVTLIIIIVVCMVVLLAVL | 2.96 | 6.11 | Vacuole |
| Syn8 | NGNCVIILVLIVVLLLLLLV | 3.00 | 61.33 | Golgi-Vacuole |
| Pep12 | VYLLIVLLVMLLFIFLIMKL | 3.04 | 63.42 | Golgi-Vacuole |
| Tlg1 | DCCIGLLIVVLIVLLVLAFIA | 3.10 | 56.57 | Golgi |
| Sso1 | CWLIVFAIIVVVVVVVVPAVV | 3.37 | 0.18 | PM |
| Sso2 | CLIICFIIFAIVVVVVVVPSVV | 3.37 | 0.29 | PM |
| Snc1 | CLALVIIILLVVIIVPI | 3.63 | 21.36 | PM-Vesicles |
| Snc2 | CLFLVVIILLVVIIVPIVV | 3.73 | 17.20 | PM-Vesicles |

**B**

| Gene | TMD | GRAVY Score | Agadir Score |
|---|---|---|---|
| Tom7 | ILTLTHNVAHYGWIPFVLYLGW | 0.87 | 0.13 |
| MAOB | PGLLRLIGLTTIFSATALGFLAHKRGL | 0.90 | 1.21 |
| BAX | TVTIFVAGVLTASLTIWKKMG | 1.16 | 0.45 |
| Tom5 | QAAYVAAFLWVSPMIWHLV | 1.28 | 0.33 |
| RHOT1 | WLRASFGATVFAVLGFAMYKALL | 1.30 | 0.65 |
| Tom6 | LYTIALNGAFFVAGVAFIQSP | 1.35 | 0.20 |
| YFL046w | VMQWLIGVCTGTFALVLAYMRLL | 1.58 | 1.54 |
| MAOA | VSGLLKIIGFSTSVTALGFVL | 1.68 | 1.09 |
| Tom22 | LAWTLTTTALLLGVPLSLSILA | 1.69 | 1.82 |
| CYB5B | WAYWILPIIGAVLLGFLY | 1.81 | 0.53 |
| RHOT2 | GLLGVVGAAVAAVLSFSLYRVLV | 1.98 | 1.54 |
| Fis1 | LVGMAIVGGMALGVAGLAGLI | 2.16 | 0.25 |
| OMP25 | IPIFMVLVPVFALTMVAAWAF | 2.29 | 0.75 |
| MAVS | GALWLQVAVTGVLVVTLLVVL | 2.34 | 0.28 |
| Fis1 | VVVAGGVLAGAVAVASFFL | 2.39 | 0.28 |

**C**

| Gene | TMD | GRAVY Score | Agadir Score |
|---|---|---|---|
| Bos1 | LVFWIALILLIIGIYYVL | 2.81 | 13.19 |
| 2AG | LVAGIALILLIIGIYYVL | 2.78 | 11.48 |
| 3AG | LVAGIALIGLIIGIYYVL | 2.55 | 0.77 |
| 4AG | LVAGIALIGAIIGIYYVL | 2.44 | 0.48 |
| 5AG | LVAGIALIGAIIGAYYVL | 2.29 | 0.46 |
| 6AG | LVAGGALIGAIIGAYYVL | 2.02 | 0.27 |
| Fis1 | VVVAGGVLAGAVAVASFFL | 2.39 | 0.28 |

GRAVY Scale: 0, 0.5, 1, 1.5, 2, 2.5, 3, 3.5, 4

Agadir Scale: 0, 10, 20, 30, 40, 50, 60, 70, 80

**Figure 1.** Sequences and properties of the TMDs of ER (**A**) and mitochondrial (**B**) TAs, as well as the model substrates used in this study (**C**). Genes highlighted in black in panel (**A**) are previously established TA substrates for the GET pathway. Sequences of TMDs were obtained from UniProt (http://www.uniprot.org)(*UniprotUniProt Consortium, 2015*). Grand Average of Hydropathy (GRAVY) scores were calculated using the GRAVY calculator (http://www.gravy-calculator.de) and are color ramped as indicated by the scale bar. Agadir scores (% helical content) was calculated at pH 7.5, 298 K, ionic strength 0.15 M using the Agadir prediction algorithm based on helix/coil transition theory (http://agadir.crg.es) (*Muñoz and Serrano, 1997*), and are color ramped as indicated. We note that helix formation is highly environment-dependent and favored in apolar environments. The Agadir algorithm calculates the local helical content of the sequence in aqueous environments, and hence should be interpreted in a relative, rather than absolute sense. Abbreviations: PM, plasma membrane; NE, nuclear envelope.

residues in the Bos1 TMD with Ala and Gly, creating a set of substrates whose TMDs increasingly mimic that of Fis1p (*Figure 2A*, 2AG-6AG). We note that these variations change the hydrophobicity of the TMD (*Figure 1C*, GRAVY scores) but also alter other properties, such as helical propensity (*Figure 1C*, Agadir scores); these properties are partially distinguished by correlation analyses below. To isolate the contribution of C-terminal charges, we swapped the TMDs and CTEs of Bos1p and Fis1p (*Figure 2A*, Bos1-FisC and Fis1-BosC); we also systematically varied the number of basic residues in the CTE (*Figure 2A*, Bos1-RR, Bos1-RRRR and Fis1-RR, Fis1-RRRR).

To test if the GET pathway can select substrates based on these features and whether this selection can be recapitulated in vitro, we measured the ability of purified Get3 to target and translocate TA substrates into ER microsomes in a Δ*get3* yeast lysate (*Rome et al., 2013*; *Wang et al., 2010*) (*Figure 2B–E*). Successful translocation into the ER enables glycosylation of an opsin tag fused to the C-terminus of substrates (*Figure 2A*), providing a semi-quantitative measure for targeting and translocation efficiency (*Rome et al., 2013*). A spacer sequence between the CTE and glycosylation site (*Figure 2A*) allows efficient glycosylation of the substrate protein in the ER (*Abell et al., 2007*;

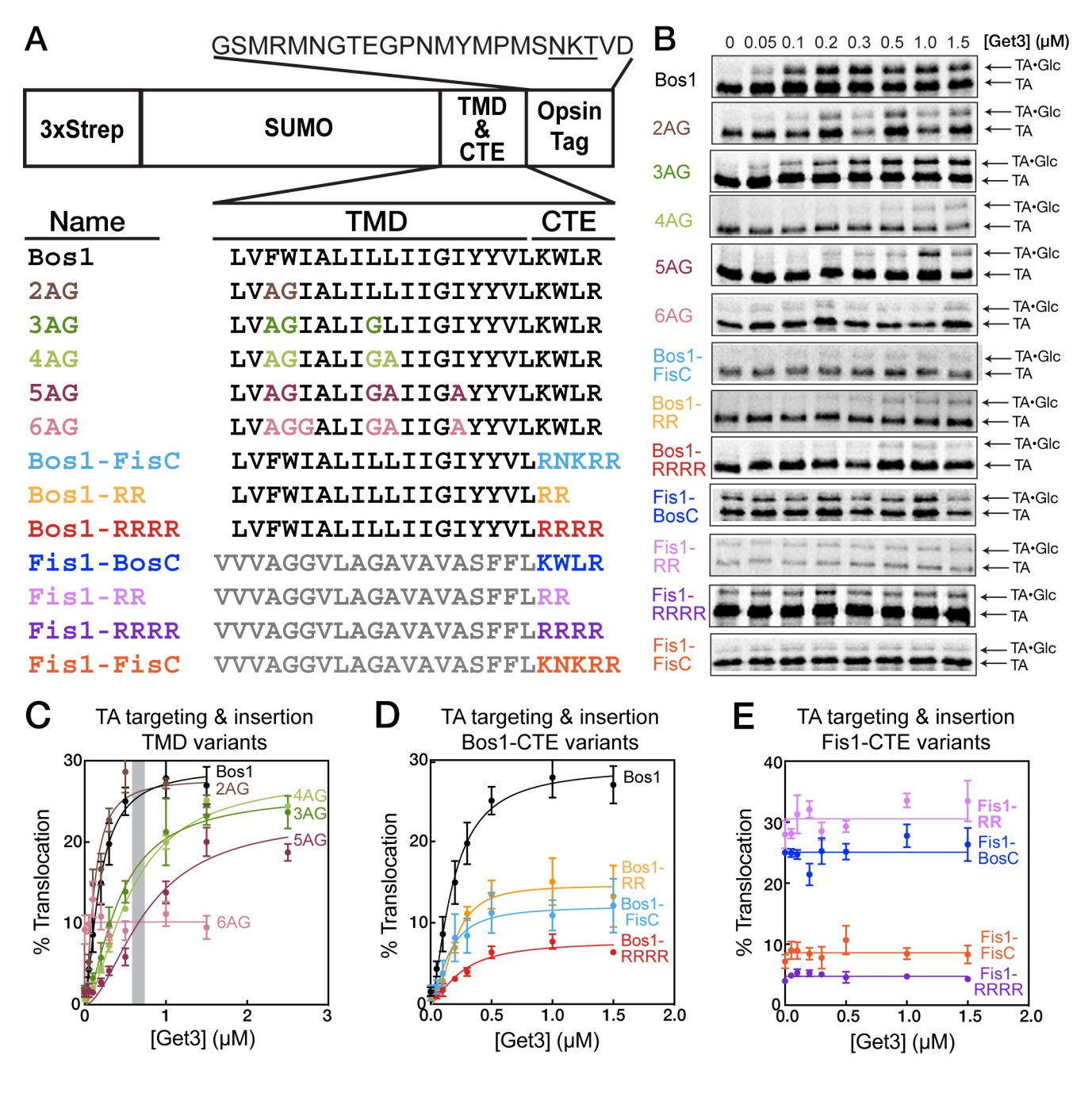

**Figure 2.** TA protein targeting and translocation into ER are sensitive to variations in both the TMD and CTE. (**A**) Nomenclature and schematic of the model substrates used in this work. The oligosaccharide transferase recognition site in the opsin tag is underlined. (**B**) Overall targeting and translocation of model TA substrates into ER microsomes. TAs are translated in a Δ*get3* yeast lysate and presented to Δ*get3* microsomes in the presence of indicated concentrations of purified Get3. (**C–E**) Quantification of the translocation of TAs for TMD variants (**C**), CTE variants with Bos1-TMD (**D**), and CTE variants with Fis1-TMD (**E**). The grey bar in (**C**) denotes the cellular concentration of Get3 in yeast (*Ghaemmaghami et al., 2003*). With the exception of 6AG, the data in panels C and D were fit to *Equation 1*, and the derived parameters are summarized in *Table 1*. The data for 6AG (panel C), Fis1-BosC, Fis1-FisC, Fis1-RR, and Fis1-RRRR (panel E) were fit to horizontal lines with y = 10 ± 0.5, y = 25 ± 0.6, y = 9 ± 0.5, y = 31 ± 0.7, and y = 5 ± 0.2%, respectively. Values are reported as mean ± S.E.M, with n = 3–6.

**Table 1.** Summary of kinetic parameters for TA targeting and translocation in Δ*get3* lysate, derived from fits of the data in *Figure 2C–D* to *Equation 1*. Values are reported as mean ± S.E.M., with n = 3–6.

| Substrate | $K_{1/2}$ (µM) | Hill coefficient | $T_{max}$ (%) |
|---|---|---|---|
| Bos1 | 0.18 ± 0.03 | 1.5 ± 0.3 | 29 ± 2.3 |
| 2AG | 0.11 ± 0.02 | 1.6 ± 0.4 | 28 ± 1.8 |
| 3AG | 0.42 ± 0.06 | 1.5 ± 0.2 | 26 ± 1.5 |
| 4AG | 0.57 ± 0.06 | 1.6 ± 0.2 | 28 ± 1.6 |
| 5AG | 0.76 ± 0.16 | 1.8 ± 0.4 | 23 ± 1.8 |
| Bos1-FisC | 0.15 ± 0.06 | 1.6 ± 0.8 | 12 ± 2.0 |
| Bos-RR | 0.19 ± 0.04 | 2.1 ± 0.8 | 15 ± 1.4 |
| Bos-RRRR | 0.24 ± 0.07 | 1.5 ± 0.5 | 8.0 ± 1.1 |

*Nilsson and von Heijne, 1993*) and minimizes interference of oligosaccharide transferase by the CTE. Previous work also showed that the addition of four leucines to the Fis1p TMD enables efficient glycosylation of Fis1-opsin fusion protein in ER microsomes (*Beilharz et al., 2003*; *Wang et al., 2010*), indicating that the charged CTE from Fis1p does not interfere with glycosylation.

Of the TAs with altered TMDs, 2AG was translocated as efficiently as Bos1; 3AG, 4AG, and 5AG underwent Get3-dependent insertion but exhibited increasing defects, especially at physiological Get3 concentrations (*Figure 2C*, gray bar); and 6AG abolished Get3-dependent translocation (*Figure 2B,C*). Increasing the number and density of basic residues in the CTE of Bos1 (Bos1-RR, Bos1-FisC, and Bos1-RRRR) also substantially reduced Get3-dependent translocation (*Figure 2B,D*). Reciprocally, reducing the positive charges in the CTE of Fis1-FisC by replacing it with the Bos1 CTE (Fis1-BosC) or with two arginines (Fis1-RR) enhanced TA insertion into ER, whereas replacing the Fis1 CTE with four arginines (Fis1-RRRR) further reduced the level of ER insertion (*Figure 2B,E*). Thus, efficient TA targeting to the ER depends on both the TMD and positive charges in the CTE, and these dependences can be recapitulated in vitro using the tested model substrates.

Several additional inferences can be made from these data. First, the translocation defects of 3AG, 4AG, and 5AG are more pronounced at Get3 concentrations below 1 µM but can be rescued by higher amounts of Get3 to levels comparable to that of Bos1 (*Figure 2C*). In contrast, the translocation of Bos1-CTE mutants saturated at ≤0.5 µM Get3 and was not further improved by higher Get3 concentration (*Figure 2D*). This suggests that distinct mechanisms are used to reject a suboptimal TMD versus CTE of TA substrates; this hypothesis is supported by in-depth analyses below. Second, substrates containing a Fis1 TMD exhibited Get3-independent insertion into ER microsomes, and insertion was also abolished by a highly charged CTE (*Figure 2E*). Thus, substrates containing the Fis1p TMD can be targeted to the ER by alternative pathways, and importantly, a positively charged CTE serves as a general feature to reject TA proteins from the ER in both GET-dependent and GET-independent pathways.

The role of the TMD in directing proteins to the ER versus mitochondria has been extensively studied in vivo (*Borgese et al., 2007*, *2003*, *2001*; *Marty et al., 2014*; *Pedrazzini, 2009*). The ability of a basic CTE to help direct TA proteins to mitochondria has also been well documented (*Borgese et al., 2003*, *2001*; *Marty et al., 2014*). However, whether a basic CTE can in addition serve as an 'ER-avoidance' sequence in vivo is unclear. To address this question, we examined the in vivo localization of GFP fused to the TMD and CTE variants of Fis1. As previously reported (*Habib et al., 2003*), the TMD and basic CTE of Fis1 (*Figure 3*, Fis1-TMD-CTE) were sufficient to direct GFP exclusively to the OMM. Deletion of the CTE led to a loss of specific mitochondrial targeting and accumulation of Fis1-TMD in the ER, as shown by the co-localization of GFP with the ER marker Sec63 tagged with tdTomato (*Figure 3*, Fis1-TMD). We next probed the minimal charge requirement of the CTE that provides mitochondrial specificity and excludes the Fis1-TMD from the ER. To this end, we created variants of Fis1-CTE with increasing numbers of arginine residues (*Figure 3*, Fis1-TMD-nR) and quantified their co-localization with Sec63-tdTomato and mitochondrially

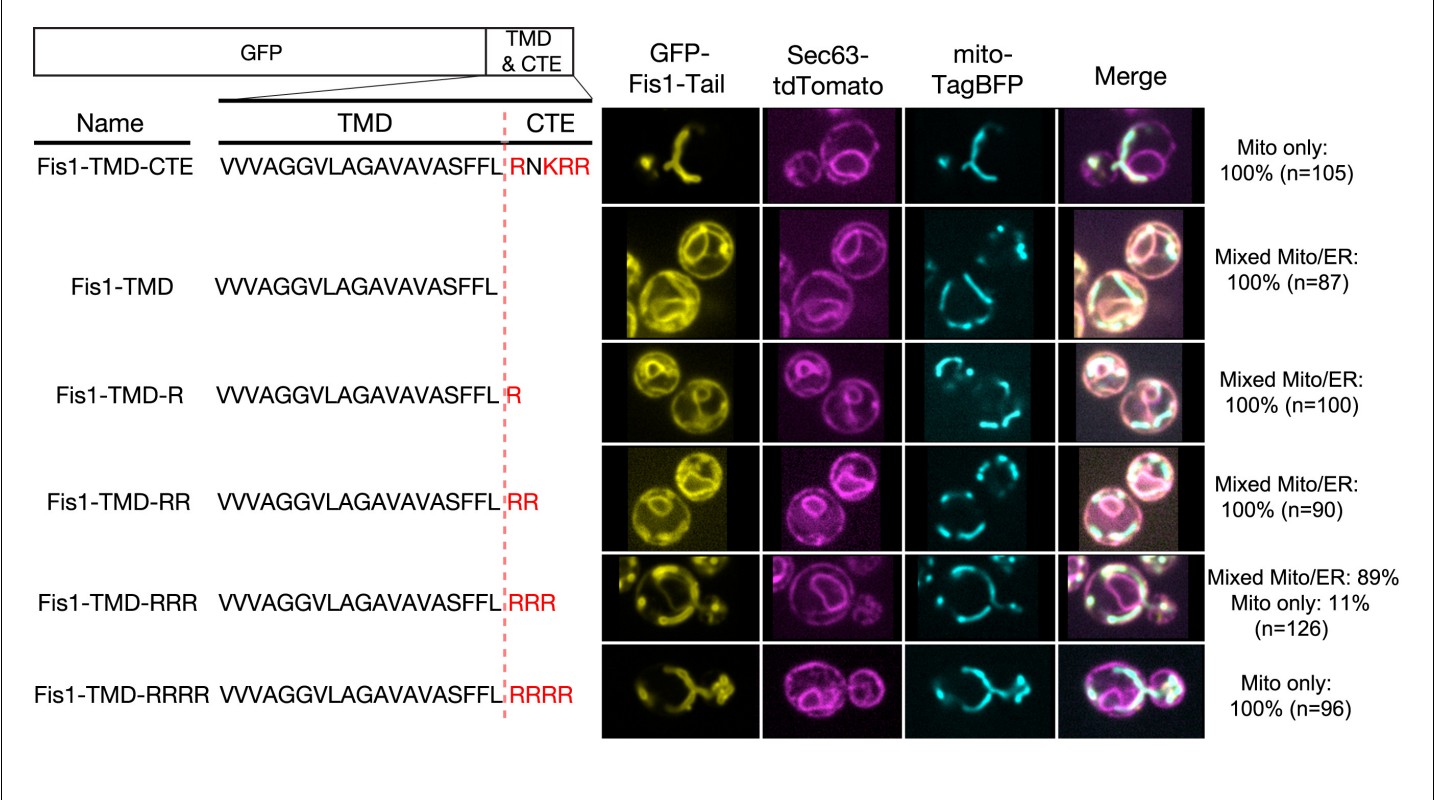

**Figure 3.** Positive charges in the CTE help reject TA proteins from ER and selectively target TAs to mitochondria. Nomenclature and schematic of the constructs used for live-cell imaging of cells expressing GFP-tagged Fis1 tail constructs are shown on the left. Basic residues in the CTE are highlighted in red. Medial focal planes are shown on the right, with ER marked by Sec63-tdTomato and mitochondria by mitochondrially targeted TagBFP. Quantification denotes % of cells in each category (mixed Mito/ER or Mito only).

targeted TagBFP (mito-TagBFP). These data showed that there is a significant reduction in the ER localization of protein when the number of positive charges on the CTE is +3 (*Figure 3*, Fis1-TMD-RRR). When the number of charges in the CTE reached +4 (*Figure 3*, Fis1-TMD-RRRR), equivalent to that of the native Fis1-CTE, specific mitochondrial localization was fully restored. Thus, in agreement with the results of the in vitro experiments (*Figure 2D,E*), a highly basic CTE plays an important role in excluding mitochondria-destined TA proteins from the ER.

## Sgt2 discriminates against TAs with suboptimal TMDs

To understand how the GET pathway selects ER-destined TA substrates, we dissected the individual molecular steps in this pathway. To this end, we adapted an *E. coli* in vitro translation system (*Goerke and Swartz, 2009*; *Jewett and Swartz, 2004*). Highly efficient translation in this lysate provides a robust source of TA proteins. Further, the lack of GET homologues in bacteria makes this lysate a bio-orthogonal system in which most of the identified molecular steps in the GET pathway can be reconstituted using purified components.

The first known step in the GET pathway is the capture of TA substrates by the co-chaperone Sgt2. To understand whether and how TA substrates are distinguished during this step, we translated TA proteins in *E. coli* lysate in the presence of $^{35}$S-methionine and His$_6$-tagged Sgt2, and analyzed the amount of TA substrate associated with Sgt2 after affinity-capture with Ni-NTA (*Figure 4A*). To provide better quantification and reduce variability, each substrate was translated and captured together with a smaller Bos1 construct lacking the 3xStrep tag (Ctrl). Both the substrate of interest and Ctrl were detected and analyzed on the same SDS-PAGE and autoradiogram (*Figure 4B*), so that the capture efficiencies of the substrates of interest were directly normalized against Ctrl. The only exceptions were the experiments with Fis1-FisC and Fis1-BosC, in which the

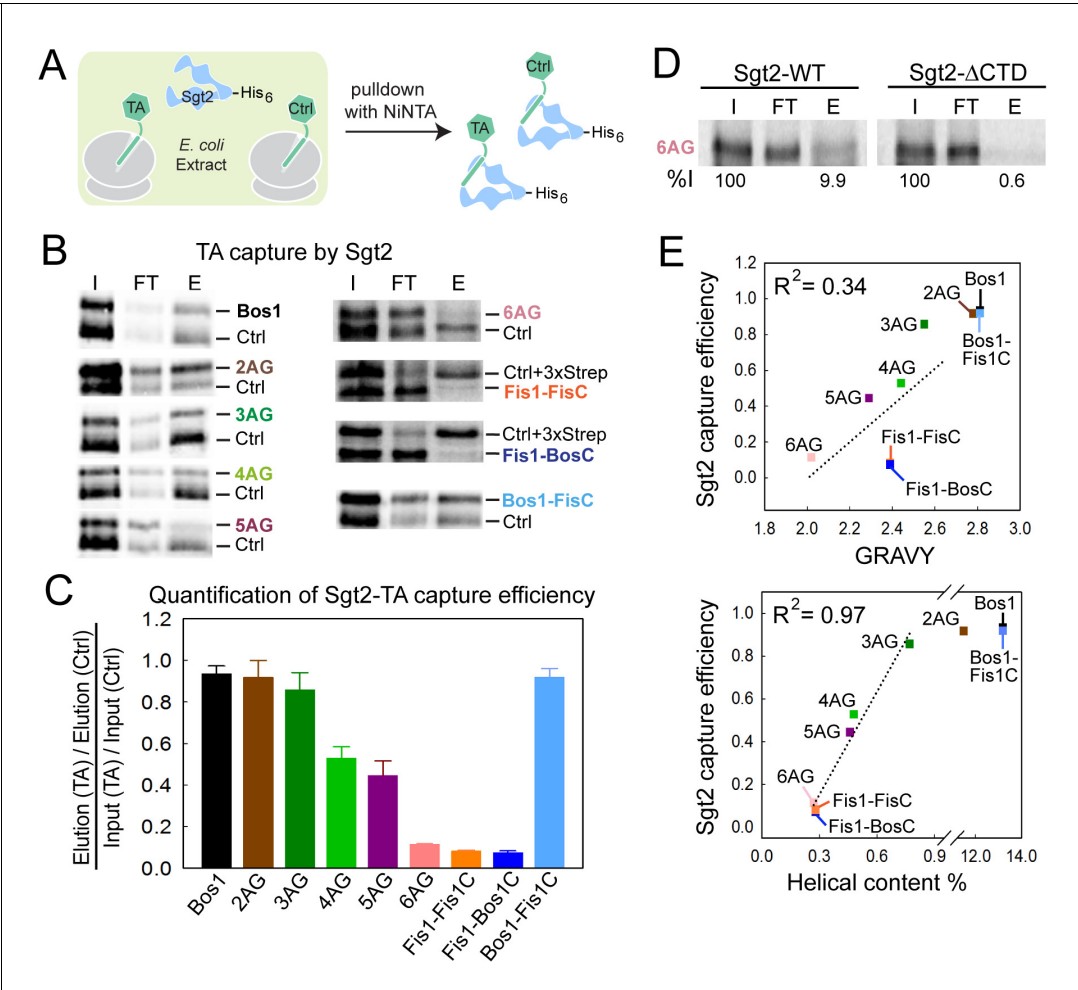

**Figure 4.** Sgt2 discriminates TAs based on features in the TMD. (**A**) Schematic of the assay to measure and compare the efficiency of TA capture by Sgt2. All TAs were translated, captured, and purified in parallel with an internal control, Bos1 (Ctrl + 3xStrep) or Bos1 lacking the N-terminal 3xStrep tag (Ctrl). (**B**) Representative autoradiograms of $^{35}$S-methionine labeled TA substrates during affinity purification of His$_6$-Sgt2•TA complexes. I, FT, and E denote input, flowthrough, and elution, respectively. (**C**) Quantification of the experiments in (**B**) and their repetitions. All values for the TA of interest were normalized against Ctrl or Ctrl + 3xStrep. Normalized TA capture efficiencies were reported as mean ± S.E.M, with n = 3–7. (**D**) Representative autoradiogram of $^{35}$S-methionine labeled 6AG during capture and purification by wildtype His$_6$-Sgt2 (left panel) or mutant His$_6$-Sgt2△CTD lacking the TA binding domain (right panel). Quantifications of substrate recovery in the elutions were indicated below the gels. (**E**) Correlation of the relative Sgt2 capture efficiencies of TA variants with the GRAVY scores (top panel) and helical content (bottom panel) of their TMDs. Substrates included in the correlation analyses are 3AG–6AG, Fis1-FisC and Fis1-BosC. The high capture efficiencies of Bos1, Bos1-FisC and 2AG (>90%) render them outside the dynamic range of this assay and were therefore not included in the correlation analysis. GRAVY scores and % helical content were from *Figure 1C*.

3xStrep tag was placed on the control (Ctrl + 3xStrep) rather than on the substrate of interest. This change was necessary to enhance the translation of Fis1-TMD containing substrates, but otherwise did not affect the outcome or interpretation of the experiment.

The results showed that within the TMD variants, Bos1, 2AG, and 3AG were captured efficiently by Sgt2; 4AG and 5AG exhibited statistically significant defects; and 6AG, Fis1-FisC, and Fis1-BosC were poorly captured by Sgt2 (*Figure 4B,C*). The small amounts of TA capture observed with 6AG, Fis1-FisC, and Fis1-BosC were reproducible. Further, control experiments with a mutant Sgt2 lacking the C-terminal TA binding domain (Sgt2△CTD) showed that the background in these experiments was ~0.6% of input, 15-fold lower than the amount of 6AG captured by wildtype Sgt2 (*Figure 4D*). In contrast to the TMD variants, Bos1-FisC, which contains a highly charged CTE, was captured with the same efficiency as Bos1 (*Figure 4B,C*). Thus, the efficiency of TA capture by Sgt2 is sensitive to variations in the TMD, but not to basic residues in the CTE.

Much discussion has focused on the role of hydrophobicity of the TMD in TA targeting to ER (*Borgese et al., 2007*, *2003*, *2001*; *Kutay et al., 1993*; *Marty et al., 2014*; *Pedrazzini, 2009*) and recognition by the GET pathway (*Mariappan et al., 2010*; *Stefanovic and Hegde, 2007*; *Wang et al., 2010*). Although hydrophobicity is an important factor, we found that it alone is insufficient to account for the data observed here: the correlation between the GRAVY scores of the TMD variants and their efficiencies of capture by Sgt2 is $R^2 = 0.34$ (*Figure 4E*, top panel). Instead, a much stronger correlation was found between the TA capture efficiency of Sgt2 and the predicted helical content of the TMD variants ($R^2 = 0.97$; *Figure 4E*, bottom panel). This suggests that a combination of hydrophobicity and helical propensity in the TMD dictates substrate recognition by Sgt2. On average, the TMDs of most OMM TA proteins are lower in helical content compared to ER-targeted TAs (*Figure 1*, Agadir Score), providing independent support that helical propensity of the TMD may serve as another distinguishing feature between ER and mitochondrial TAs.

## The Fis1 TMD is rejected prior to TA loading onto Get3

In the next step of the pathway, TA substrates are transferred from Sgt2 to Get3 with the help of the Get4/5 complex. To quantitatively understand this substrate handover event, we developed an assay based on Förster Resonance Energy Transfer (FRET). Using an *E. coli* translation lysate that harbors a pair of engineered amber suppressor tRNA and tRNA synthetase, a fluorescent unnatural amino acid, 7-hydroxycoumaryl ethylglycine (Cm), was efficiently and site-specifically incorporated into the TA substrate during translation (*Figure 5A,B*) (*Saraogi et al., 2011*; *Wang et al., 2006*). Cm was incorporated four residues upstream of the TMD of TA substrates and served as the FRET donor (denoted as TA$^{Cm}$). As the FRET acceptor, CoA-BODIPY-FL was enzymatically conjugated to ybbR-tagged Get3 via the Sfp phosphopantetheinyl transferase enzyme (*Figure 5E–G*; [*Yin et al., 2006*]). BODIPY-FL-labeled Get3 (denoted as Get3$^{BDP}$) exhibits translocation and ATPase activities similar to those of wildtype Get3 (*Figure 5H,I*).

To reconstitute TA transfer from Sgt2 to Get3, we generated and affinity purified Sgt2•TA$^{Cm}$ complexes (*Figure 5C,D*) and incubated the complex with Get4/5, ATP, and Get3 to allow TA transfer (*Figure 6A*). We observed a significant reduction in Cm fluorescence when the transfer reaction was carried out with Get3$^{BDP}$ (*Figure 6B*, purple vs. green). As a control, no change in TA$^{Cm}$ fluorescence was observed when the transfer reaction was carried out with unlabeled Get3 (*Figure 6B*, purple vs. blue), indicating the absence of environmental effects on Cm fluorescence upon TA transfer. These results demonstrate a high efficiency of both TA transfer and FRET between TA$^{Cm}$ and Get3$^{BDP}$. Consistent with the contribution of Get4/5 to TA transfer observed previously (*Mateja et al., 2015*; *Wang et al., 2010*), the presence of Get4/5 accelerated the TA transfer reaction 5–10 fold (*Figure 6C*, grey vs. black, and *Table 2*). Comparable fluorescence changes were obtained at the end of TA transfer reactions with and without Get4/5 present (*Figure 6C*); this corroborates that the observed FRET arose from loading of TA$^{Cm}$ onto Get3$^{BDP}$, rather than their being adjacent to one another in a TA•Sgt2•Get4/5•Get3 complex. Finally, rapid TA transfer was observed even when the reaction was carried out in the presence of ribosome-depleted △*get3* lysate (*Figure 6D*), suggesting that Get4/5-dependent TA transfer is robust and can withstand competition from cellular factors.

Using this FRET assay, we asked whether TA transfer from Sgt2 to Get3 is sensitive to variations in the TA substrate. We first measured FRET when the transfer reaction was allowed to reach equilibrium at varying concentrations of Get3$^{BDP}$ (*Figure 6E*). These titrations showed that the apparent equilibrium of TA transfer between Sgt2 and Get3, quantified empirically by the concentration of Get3 required for 50% transfer ($K_{1/2}$), ranged from 3.7–10 nM for various substrates in the presence of 150 nM Sgt2 (*Figure 6E,F* and *Table 2*). This indicates that, although both Sgt2 and Get3 bind TA substrates, the equilibrium for interaction with TA is 15–40 fold in favor of Get3. The modest variation of this transfer equilibrium (<3 fold) among Bos1 and 2AG–5AG indicates that the relative preference of Get3 for this set of TA variants closely parallels those of Sgt2, such that TA transfer from Sgt2 to Get3 is largely isoenergetic among these substrates. Real-time measurements of the TA transfer reaction showed that, although the transfer kinetics is complex and exhibited bi-phasic behavior (*Figure 6C,D* and *Table 2*), Bos1 and 2AG–5AG were all rapidly transferred from Sgt2 to Get3, with reaction halftimes ($t_{1/2}$) of 5–17 s (*Figure 6G*). Analogously, Bos1-FisC, which contains a charged CTE, exhibited similar transfer equilibrium and kinetics compared to those of Bos1 (*Figure 6E–G*, *blue*). Thus, substrate handover from Sgt2 to Get3 is only modestly sensitive to

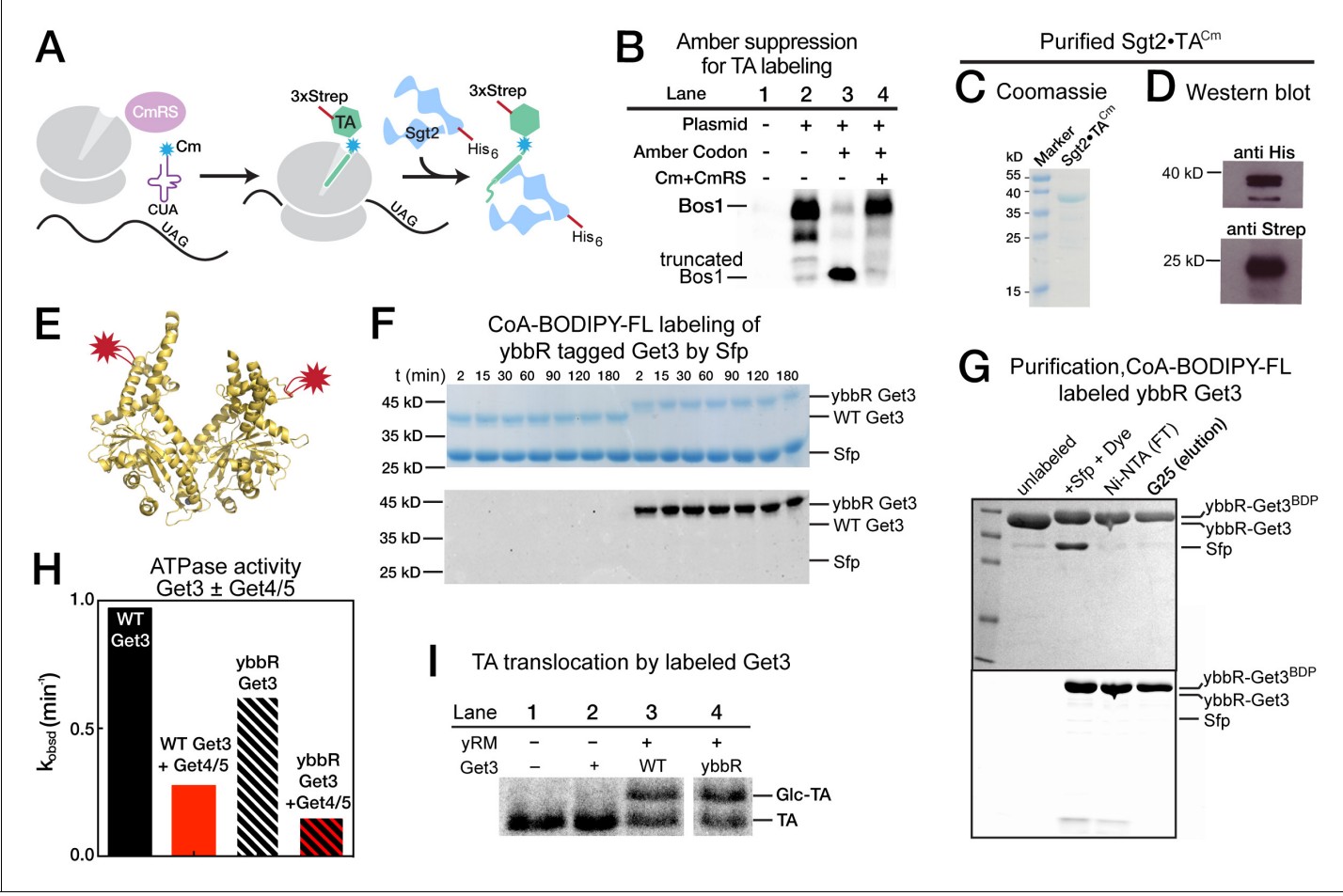

**Figure 5.** Labeling of TA and Get3. (**A**) Scheme of TA labeling with the fluorescent amino acid Cm using amber suppression technology in *E. coli* lysate. (**B**) Autoradiogram showing amber suppression efficiency. Bos1 was translated in the presence of $^{35}$S-methionine without (lane 2) or with (lanes 3,4) an amber codon four residues N-terminal to the TMD, in the absence (lane 3) and presence (lane 4) of Cm and Cm synthetase. (**C**, **D**) Coomassie-stained SDS-PAGE gel (**C**) and western blot (**D**) of a purified Sgt2•TA$^{Cm}$ complex. Sgt2 was His$_6$ tagged and TA was 3xStrep-tagged. (**E**) Location of the ybbR tag (for labeling) on the structure of Get3 (PDB: 3H84). (**F**) Coomassie-stained (top) and in-gel fluorescence (bottom) of Sfp-mediated conjugation of CoA-BODIPY-FL to the Get3 ybbR tag. (**G**) Purification of labeled ybbR-Get3. Top, coomassie-stain; bottom, in-gel fluorescence. (**H**) ATPase assays of wildtype and labeled ybbR-Get3. (**I**) Representative autoradiograms for the translocation reactions with wildtype and labeled ybbR-Get3.

changes in this set of TMD variants and insensitive to the positive charges in the CTE of TA substrates.

Notably, TA transfer to Get3 was not observed for Fis1-FisC (*Figure 6C*, *orange*). The Fis1-TMD is responsible for the absence of observed transfer, as Fis1-BosC was also strongly discriminated during this reaction (*Figure 6C*, *navy*), whereas Bos1-FisC was not (*Figure 6E–G*, *blue*). During the transfer experiment, the TA proteins could partition between multiple routes including transfer onto Get3 (observed by the FRET assay), dissociation from Sgt2, and potentially other off-pathway processes not observed in the FRET assay. The absence of any observed FRET in these measurements indicates highly unfavorable partitioning of Fis1-BosC and Fis1-FisC into the transfer reaction compared to the alternative routes. It is unclear whether the absence of observed transfer arises exclusively from the rapid dissociation of these substrates from Sgt2, or whether substrate transfer to Get3 imposes a higher barrier for TA proteins containing suboptimal TMDs. Nevertheless, these observations are consistent with the Get3-independent targeting of substrates containing the Fis1-TMD in complete targeting and translocation assays (*Figure 2E*) and indicates that these substrates are completely rejected at steps upstream of TA loading onto Get3.

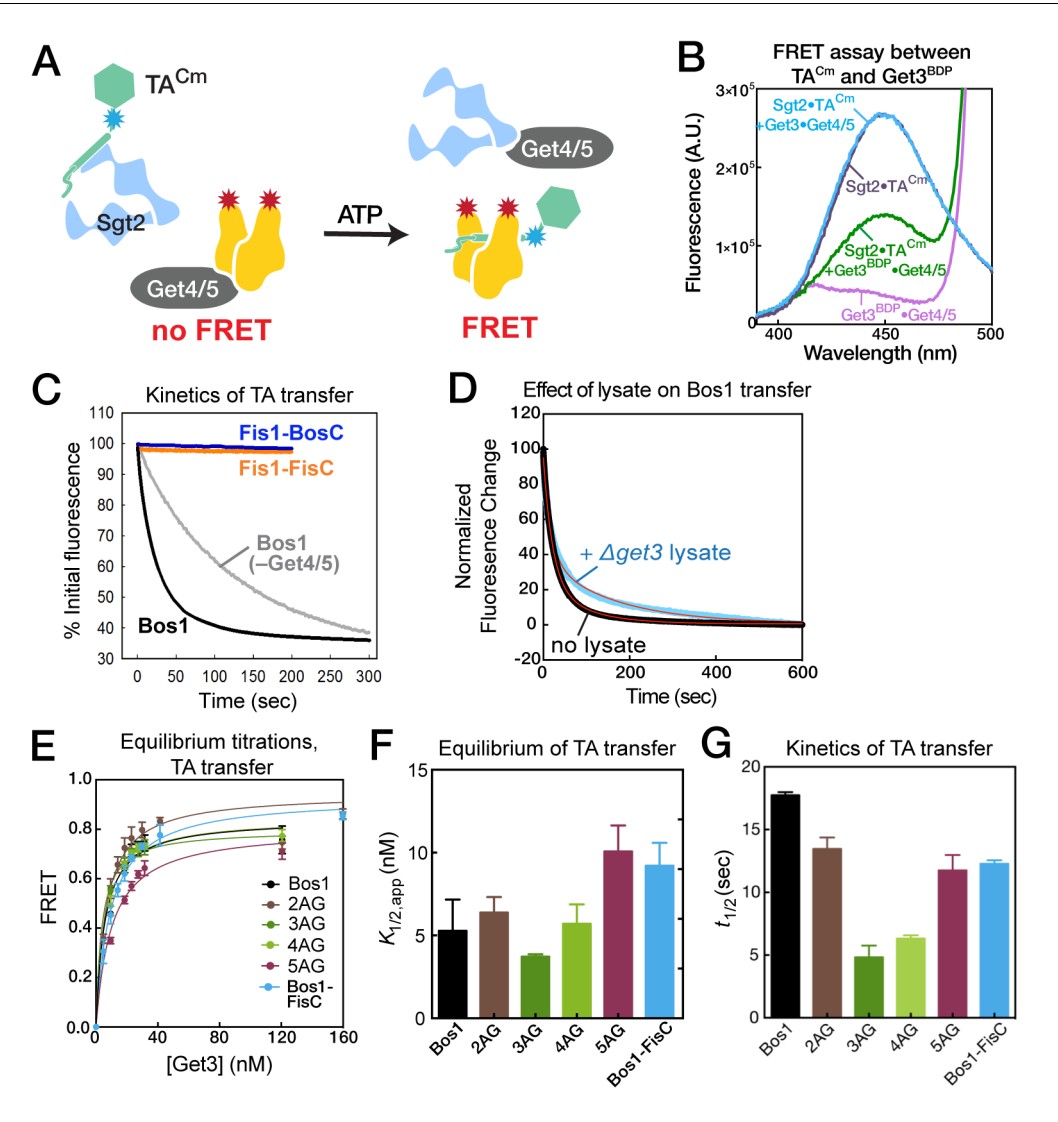

**Figure 6.** Equilibrium and kinetics of TA transfer from Sgt2 to Get3. (**A**) Scheme of the TA transfer reaction and the FRET assay to monitor this transfer. Purified Sgt2•TA$^{Cm}$ complexes were presented to the Get3$^{BDP}$•Get4/5 complex in the presence of ATP. Loading of TA$^{Cm}$ onto Get3$^{BDP}$ results in FRET between the dye pair. (**B**) Fluorescence emission spectra for purified Sgt2•TA$^{Cm}$ complex (purple; donor fluorescence), Sgt2•TA$^{Cm}$ incubated with unlabeled Get3, Get4/5 and ATP (blue; donor fluorescence corrected for environmental sensitivity), Get3$^{BDP}$ and Get4/5 (magenta; acceptor fluorescence), and Sgt2•TA$^{Cm}$ complex incubated with Get3$^{BDP}$, Get4/5 and ATP (green; donor fluorescence in the presence of acceptor). (**C**) Representative time courses of the transfer of Bos1 with (black) and without Get4/5 (gray) present, and transfer of Fis1-FisC (*orange*) and Fis1-BosC (*navy*) with Get4/5 present. The data were fit to *Equation 4*, and the derived kinetic parameters are summarized in panel G and in *Table 2*. (**D**) Transfer of Bos1 from Sgt2 to Get3 in the presence (blue) and absence (black) of a ribosome depleted △*get3* lysate. Red lines are fits of the data to *Equation 4*, and the derived values are summarized in *Table 2*. (**E**, **F**) Equilibrium measurements of TA transfer reactions as a function of Get3 concentration. All reactions used 50 nM Sgt2•TA$^{Cm}$ complexes supplemented to a final concentration of 150 nM Sgt2, 150 nM Get4/5, 2 mM ATP, and indicated concentrations of Get3$^{BDP}$. The data in panel E were fit to *Equation 3*, and the values are reported in panel F and *Table 2*. (**G**) Summary of the half-times of TA transfer reactions for various TA proteins.

**Table 2.** Summary of the equilibrium and kinetic parameters for TA transfer from Sgt2 to Get3, derived from the data in **Figure 6**. Refer to **Equations 3 and 4** for definitions of the parameters. Values are reported as mean ± S.E.M., with n ≥ 2. N.D., not determined.

| Substrate | $K_{1/2}$ (nM) | FRET endpoint | $k_{fast}$ ($10^{-3}$ s$^{-1}$) | $\triangle F_{fast}$ | $k_{slow}$ ($10^{-3}$ s$^{-1}$) | $\triangle F_{slow}$ | $t_{1/2}$ (s) |
|---|---|---|---|---|---|---|---|
| Bos1 (−Get4/5) | N.D. | N.D. | 9.4 ± 0.07 | 0.76 ±0.03 | 3.0 ± 0.1 | 0.24 ±0.03 | 98 ± 8.7 |
| Bos1 | 5.2 ± 1.0 | 0.84 ± 0.03 | 47 ± 0.4 | 0.75 ±0.01 | 11 ± 0.4 | 0.25 ±0.01 | 17.7 ± 0.2 |
| 2AG | 6.4 ± 0.5 | 0.94 ± 0.02 | 77 ± 1 | 0.62 ±0.01 | 7.0 ± 0.1 | 0.38 ±0.01 | 13.5 ± 0.5 |
| 3AG | 3.7 ± 0.1 | 0.80 ± 0.01 | 164 ± 2 | 0.77 ±0.01 | 17 ± 0.8 | 0.23 ±0.01 | 4.8 ± 0.5 |
| 4AG | 5.7 ± 0.6 | 0.84 ± 0.02 | 183 ± 9 | 0.54 ±0.01 | 33 ± 0.2 | 0.46 ±0.01 | 6.3 ± 0.1 |
| 5AG | 10 ± 1 | 0.80 ± 0.02 | 151 ± 9 | 0.43 ±0.01 | 23 ± 0.8 | 0.57 ±0.01 | 11.8 ± 0.7 |
| Bos1FisC | 9.2 ± 0.8 | 0.93 ± 0.02 | 63 ± 0.3 | 0.83 ±0.01 | 9.0 ± 0.1 | 0.17 ±0.01 | 12.3 ± 0.2 |

## C-terminal basic residues slow TA insertion into ER

In the last stage of the GET pathway, the Get3•TA complex is targeted to the Get1/2 receptor complex, and the TA substrate is inserted into the ER membrane. To reconstitute this step, we generated $^{35}$S-methionine labeled TAs during in vitro translation in the presence of Sgt2, Get4/5, and His$_6$-tagged Get3, and affinity purified Get3•TA complexes using Ni-NTA. Purified complexes were presented to ER microsomes derived from Δget3 yeast, and the efficiency of targeting and insertion was assessed by glycosylation (**Figure 7A**). Measurement of the reaction time courses showed that 2AG was targeted and inserted with similar kinetics and efficiency as Bos1 (**Figure 7B,C**). Compared to Bos1, the insertion of 3AG–6AG were also rapid, but 5AG showed a small reduction in insertion efficiency at steady state, and 6AG showed a marked reduction in the level of insertion at steady state (**Figure 7C,E**). On the other hand, Bos1-FisC, Bos1-RR, and Bos1-RRRR exhibited significantly slower rates to reach steady state compared to Bos1 (**Figure 7C,E**), mirroring their translocation defects observed in the complete yeast lysate. We note that although the absolute insertion efficiency of each substrate could vary with different yRM preparations, the *change* in TA insertion kinetics and steady-state level caused by the TMD and CTE variants are consistent across different yRM preps (**Figure 7C,E** and **Table 3**).

Further analysis of the kinetics of TA insertion strongly suggests that the observed translocation defects of the TMD and CTE variants arise from different mechanisms. During an observed insertion reaction, the productive targeting and insertion processes (**Figure 7F**, $k_{insert}$) must compete with nonproductive events ($k_{nonproductive}$) including the reversal of this reaction and more likely, disassembly of the Get3•TA complex ($k_{dis}$) that could lead to aggregation of the TA substrate ($k_{agg}$). The observed rate constant of the insertion reaction ($k_{obsd}$) is the sum of the rate constants for insertion from the Get3/TA complex and competing nonproductive reactions, and the level of insertion at steady state is determined by the relative rates of the insertion reaction compared to the nonproductive processes (**Equations 6–7** in Materials and methods). Dissection of the observed reaction kinetics into these components showed that the observed translocation defects of 5AG and 6AG arise primarily from a faster rate of nonproductive reactions (**Figure 7G–H**, open bars and **Table 3**, $k_{nonproductive}$). In contrast, the observed defects of Bos1-FisC, Bos1-RR, and Bos1-RRRR arise primarily from a slower rate of productive insertion into the ER (**Figure 7G–H**, solid bars and **Table 3**, $k_{insert}$).

We considered two alternative possibilities that could contribute to the slower observed insertion of the CTE variants. First, the Get3•TA complexes were purified via the affinity tag on Get3 for the experiments above; the presence of excess free Get3 could trap the Get4/5 complex during purification. The Get3•TA complex must dissociate from Get4/5 to interact with the Get1/2 receptors at the ER during the observed insertion reaction (**Gristick et al., 2014**; **Rome et al., 2014**), and this

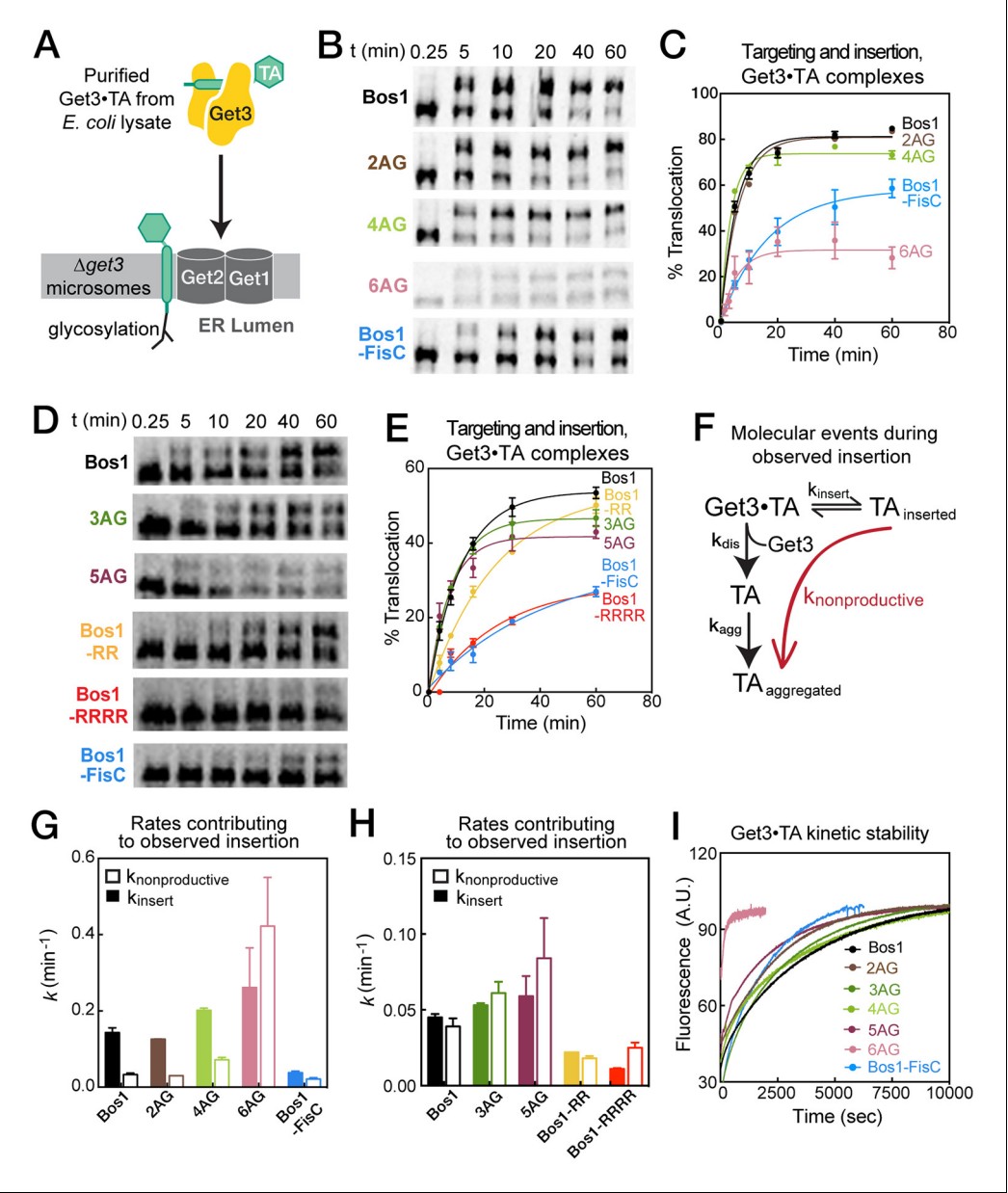

**Figure 7.** Basic residues in the CTE slow targeting and insertion of the Get3•TA complex into the ER membrane. (A) Schematic of the reaction assayed in *Figure 7*. Get3•TA complexes generated and purified from *E. coli* lysate were presented to △*get3* microsomes. Insertion was monitored by glycosylation of the opsin tag on TA substrates. (B, D) Representative autoradiograms of insertion reactions. The reactions in panels B and D used two different yRM preps. (C, E) Quantification of the insertion reactions shown in panels B and D, respectively. The data were fit to *Equation 5*, and the values obtained are summarized in *Table 3*. (F) Schematic of the molecular events during the insertion reaction. Productive TA insertion ($k_{insert}$) competes with nonproductive processes (collectively termed $k_{nonproductive}$), including reversal of the targeting reaction, disassembly of the Get3•TA complex ($k_{dis}$) and aggregation of TA substrates ($k_{agg}$). (G–H) Summary of the rate constants of competing events, defined in panel F and *Equations 6–7*, that contribute to the observed rate constants and efficiencies of the targeting/insertion reactions in panels C and E, respectively. See also *Table 3* for a summary of the values of $k_{nonproductive}$ and $k_{insert}$. (I) The kinetic stability of Get3•TA complexes ($k_{dis}$ defined in panel F) measured by pulse-chase experiments as described in the Materials and methods. The data were fit to *Equation 8* and the obtained $k_{dis}$ values are summarized in *Table 3*.

**Table 3.** Summary of rate constants for TA targeting and insertion from the Get3•TA complex, derived from the data in **Figure 7**. Refer to **Equations 5-8** for definitions of the parameters. The thick line separates the quantifications derived from the data in **Figure 7B and D**, which used two different yRM preps. Values are reported as mean ± S.E.M., with n ≥ 2. N.D., not determined.

| Substrate | $k_{obsd}$ (min$^{-1}$) | Translocation end point (%) | $k_{insert}$ (min$^{-1}$) | $k_{nonproductive}$ (min$^{-1}$) | $k_{dis}$ (min$^{-1}$) |
|---|---|---|---|---|---|
| Bos1 | 0.18 ± 0.02 | 81.2 ± 0.5 | 0.14 ±0.01 | 0.033 ± 0.004 | 0.018 ± 0.001 |
| 2AG | 0.16 ± 0.01 | 80.8 ± 0.2 | 0.13 ±0.01 | 0.030 ± 0.001 | 0.023 ± 0.001 |
| 4AG | 0.27 ± 0.01 | 74 ± 2 | 0.20 ±0.01 | 0.072 ± 0.006 | 0.017 ± 0.001 |
| 6AG | 0.70 ± 0.20 | 36 ± 4 | 0.30 ±0.10 | 0.40 ± 0.10 | 0.20 ± 0.0025 |
| Bos1-FisC | 0.059 ± 0.01 | 58.4 ± 4.6 | 0.038 ±0.010 | 0.020 ± 0.003 | 0.029 ± 0.001 |
| Bos1 | 0.085 ± 0.008 | 53.8 ± 2.1 | 0.045 ±0.002 | 0.039 ± 0.005 | 0.018 ± 0.001 |
| 3AG | 0.11 ± 0.01 | 46.7 ± 2.4 | 0.053 ±0.001 | 0.060 ± 0.008 | 0.026 ± 0.001 |
| 5AG | 0.14 ± 0.02 | 41.9 ± 2.3 | 0.060 ±0.010 | 0.080 ± 0.030 | 0.034 ± 0.001 |
| Bos1-RR | 0.040 ± 0.002 | 54.8 ± 2.1 | 0.022 ±0.001 | 0.018 ± 0.002 | N.D. |
| Bos1-RRRR | 0.036 ± 0.004 | 30.1 ± 1.7 | 0.011 ±0.001 | 0.025 ± 0.003 | N.D. |
| Bos1-FisC | 0.023 ± 0.006 | 34.9 ± 1.9 | 0.009 ±0.001 | 0.020 ± 0.005 | 0.029 ± 0.001 |

dissociation could be slower with the charged CTE variants compared to Bos1. To address this possibility, we re-purified Get3•TA complexes for Bos1, Bos1-FisC and Bos1-RRRR via the 3xStrep tag on the TA substrate. The Get3•TA complexes purified through this procedure were free of Get4/5 (**Figure 8A**). Targeting and insertion reactions with these complexes showed that the observed insertion reactions with Bos1-FisC and Bos1-RRRR remained significantly slower than those of Bos1 (**Figure 8B–D**) even without Get4/5 present. Second, ATP hydrolysis by Get3 precedes the disassembly of Get3•TA complex and insertion of TA by Get1/2 (**Rome et al., 2014, 2013**; **Wang et al., 2011**); as the TA substrate stimulates the ATPase activity of Get3 (**Rome et al., 2013**), the CTE variants could be defective in this ATPase stimulation. To rule out this possibility, we tested the targeting and insertion of Get3•TA complexes without ATP present. Although the absence of ATP reduced the overall amount of insertion, as anticipated from the role of ATP in enabling efficient Get3 turnover for multiple rounds of targeting (**Mariappan et al., 2011**; **Rome et al., 2014**; **Stefer et al., 2011**; **Wang et al., 2011**), the targeting and insertion of Bos1-FisC remained significantly slower than those of Bos1 (**Figure 8C versus E**). Together, these results show that the targeting and insertion of TA at the ER membrane provides the main selection mechanism by which substrates with charged CTE are rejected.

A likely source of the nonproductive reactions is the disassembly of the Get3•TA complex, which can lead to TA aggregation and/or misfolding. To test whether this was the case, we measured the kinetic stabilities of different Get3•TA complexes. We generated and purified Get3•TA complexes as for the insertion reactions, except that TA and Get3 were labeled with Cm and BODIPY-FL, respectively. The disassembly rates for the Get3•TA complexes were measured using pulse-chase experiments and monitored by loss of FRET between TA$^{Cm}$ and Get3$^{BDP}$. The results showed that, with the exception of 6AG, dissociation of the Get3•TA complexes is slow, with rate constants of ~0.018–0.034 min$^{-1}$ (or life time of 30–60 min) for most of the substrates tested (**Figure 7I** and **Table 3**, $k_{dis}$). Bos1-FisC, which contains the basic CTE, dissociated from Get3 with a rate constant similar to those of Bos1 and 2AG. In contrast, 6AG dissociated from Get3 more quickly, with a rate constant

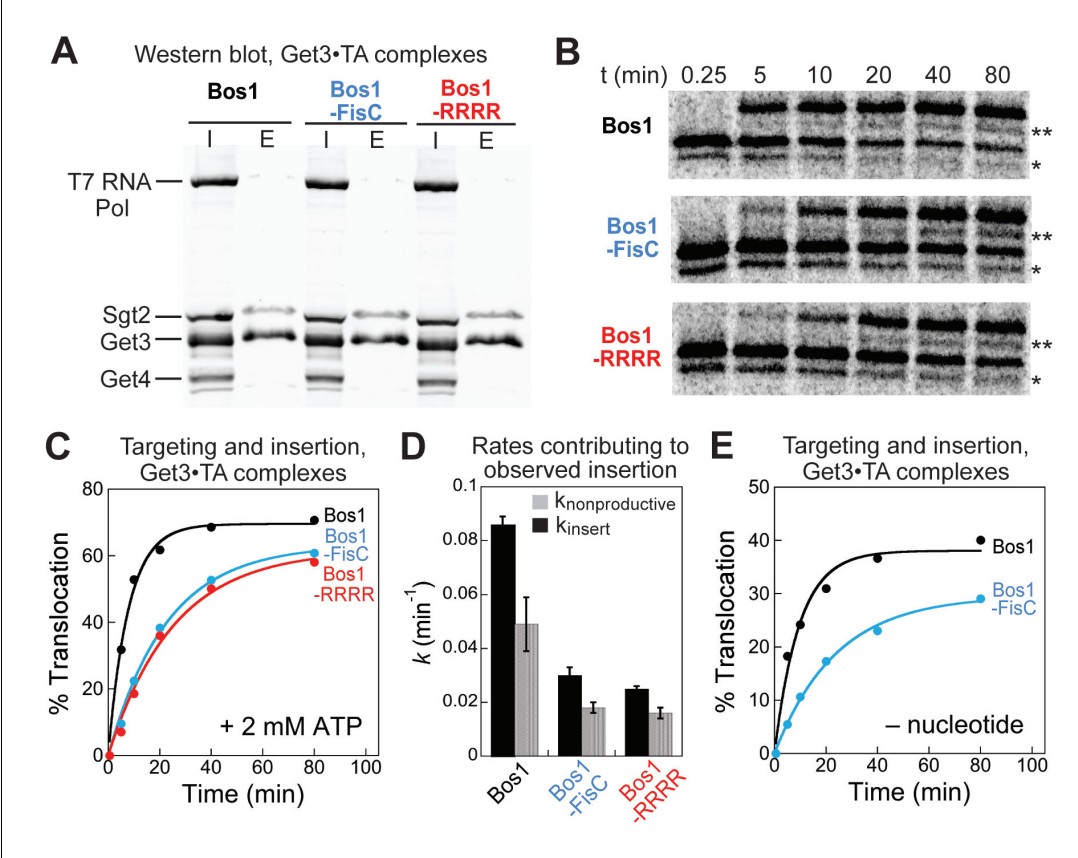

**Figure 8.** Defects of CTE variants in targeting and insertion cannot be attributed to Get4/5 or ATP hydrolysis. (A) SDS-PAGE and western blot (against His$_6$) analysis of purified Get3•TA complexes. Get3•TA complexes were assembled during in vitro translation in S30 extract in the presence of Sgt2, Get4/5 and Get3, and purified via the 3xStrep-tag on the TA protein as described in the Materials and methods. T7 RNA polymerase, Sgt2, Get3, and Get4 were His$_6$-tagged. I denotes input onto the Strep-Tactin resin, E denotes elution. (B) Representative autoradiograms of insertion reactions. The asterisk (*) denotes a truncated translation product of the TA, which was inserted and glycosylated (**) with similar efficiencies as the full-length translation product. (C) Quantification of the insertion reactions in panel B. The data were fit to *Equation 5*, which (together with replicates) gave observed insertion rate constants ($k_{obsd}$) of 0.14 ± 0.01, 0.048 ± 0.006, and 0.041 ± 0.003 min$^{-1}$, and translocation end points (T) of 64 ± 8, 63 ± 1, and 61 ± 2% for Bos1, Bos1-FisC, and Bos1-RRRR, respectively. (D) Summary of the rate constants of competing molecular events, defined in *Figure 7F* and *Equations 6–7*, that contribute to the observed targeting/insertion reactions in panel C. Values represent mean ± S.E.M., with n = 2. (E) Targeting and insertion of Get3•Bos1 and Get3•Bos1-FisC complexes in the absence of nucleotides.

of 0.20 min$^{-1}$. The measured rate constants of Get3•TA complex disassembly were consistent with those of nonproductive reactions calculated from the translocation data (*Table 3*). This provides independent evidence that, once TA proteins are loaded onto Get3, substrates with suboptimal TMDs and CTEs are rejected during the subsequent targeting and translocation via different mechanisms.

## Discussion

Accurate protein localization is essential for cells to establish and maintain compartmentalization. TA proteins, with a TMD near the C-terminus as their only defining feature, pose special challenges for protein targeting pathways that need to sort these proteins to the correct cellular membranes. In this work, systematic analyses show that the GET pathway uses multiple mechanisms to recognize distinct physicochemical features on the TA protein and select for the correct substrates. These results also suggest new roles of the co-chaperone Sgt2 and rationalize, in part, the chemical logic for the series of substrate handover events in this targeting pathway.

During initial entry into the GET pathway, capture of TA proteins by Sgt2 provides the first selection filter to discriminate substrates with suboptimal TMDs (*Figure 9A*, Step 1). We found here that TA substrates with increasing number of Ala/Gly replacements in the TMDs form less stable complexes with Sgt2, consistent with previous work showing that addition of four leucines to the Fis1-

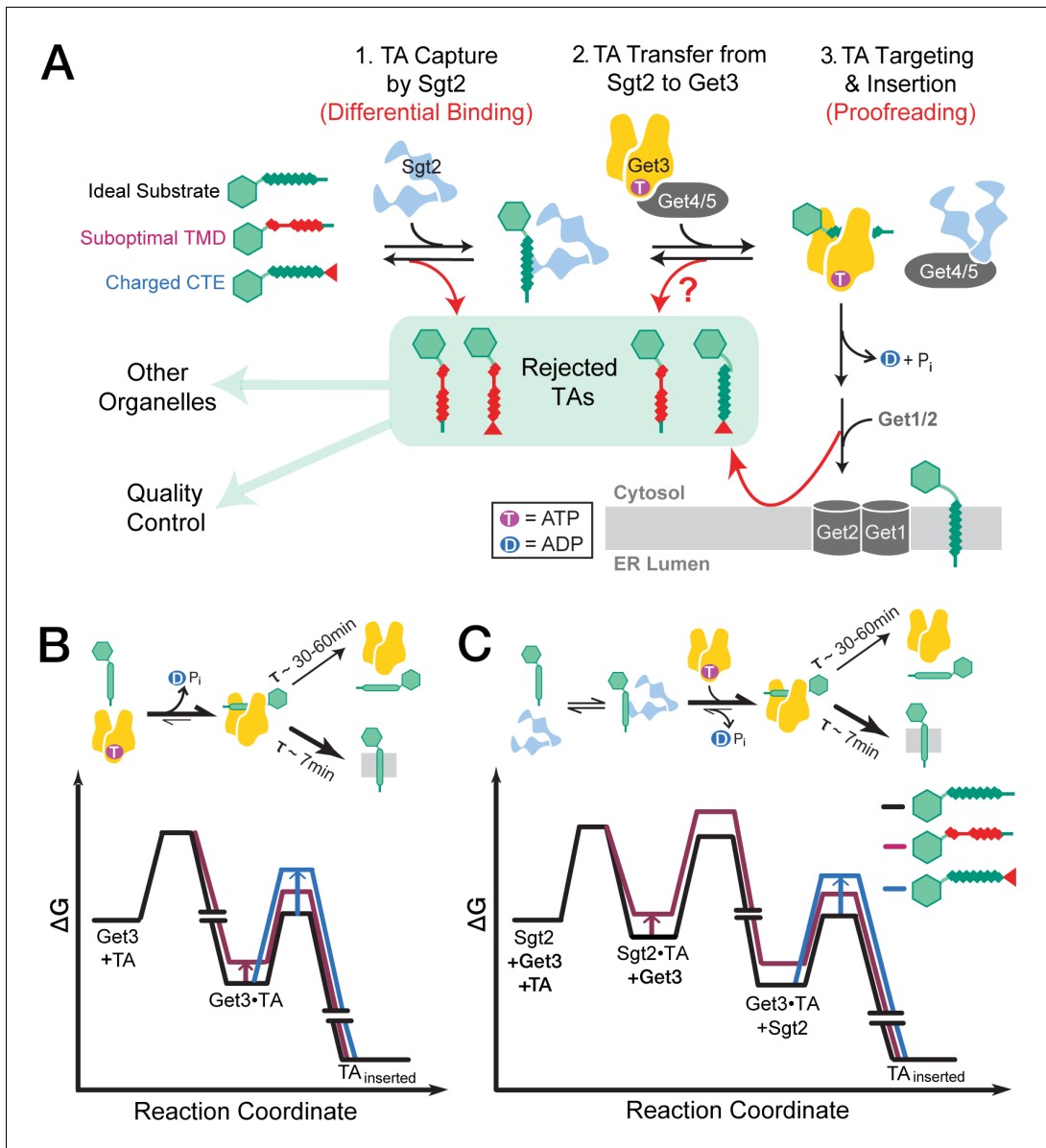

**Figure 9.** The GET pathway selects TA substrates using multiple mechanisms. (**A**) Model of sequential TA selection by the GET pathway. Step 1: TAs are captured by Sgt2. TA substrates with less suboptimal TMDs (red coil) are less favorably captured. Step 2: Get4/5 mediates TA transfer from Sgt2 to ATP-bound Get3. The question mark indicates that it is unclear whether the TA transfer step selectively poses a higher barrier for substrates containing suboptimal TMDs. Step 3: after hydrolyzing ATP, the Get3•TA complex associates with the Get1/2 membrane receptors, and the latter facilitates TA insertion. TAs enriched in basic residues at the extreme C-termini (red tail) are discriminated at this step. (**B–C**) Free energy profiles for TA capture and insertion without (**B**) and with (**C**) Sgt2 in the GET pathway. The free energy profiles for a good substrate (Bos1) are indicated in black, for a substrate with a suboptimal TMD (5AG) are in maroon, and those for a substrate with a charged CTE (Bos1-FisC and Bos1-RRRR) are in blue. The off-pathway branch in which a TA dissociates from Get3 and aggregates is not shown for simplicity.

TMD confers strong interaction with Sgt2 (*Wang et al., 2010*). The strong correlation between the TA capture efficiency of Sgt2 and the helical content of the TMD variants tested here was unexpected and suggests that overall hydrophobicity is not the sole determinant for recognition by Sgt2. Although a structure of the TA binding domain of Sgt2 is not available, these data suggest that this domain might be able to recognize secondary structure features such as a TMD helix. The overall lower helical content of the TMDs of OMM TAs compared to ER-targeted TAs (*Figure 1*) provides further support that recognition based on helical propensity could provide an effective mechanism to distinguish between ER and mitochondrial TAs. On the other hand, Sgt2 is insensitive to enrichment of basic residues at the extreme C-terminus that characterizes some mitochondrial TAs, suggesting that this property must be detected by other mechanisms.

The handover of TA proteins from Sgt2 to the targeting factor Get3 (*Figure 9A*, Step 2) appears to be isoenergetic amongst the TA variants examined here. The modest difference observed for most of the TMD variants during this transfer reaction indicates that the relative preference of Get3 for these variants closely parallels those of Sgt2. The basic CTE from Fis1p was also well tolerated during this handover, suggesting that analogous to Sgt2, Get3 does not recognize the positive charges in the CTE. It is unclear whether the lack of observed transfer for substrates containing the Fis1 TMD could be attributed solely to the lower kinetic stability of the Sgt2•TA complex with these substrates, or whether the TA handover event poses an additional selection filter for substrates with the Fis1-TMD. Nevertheless, the combination of the results from the Sgt2-TA capture and TA transfer experiments shows that substrates containing the Fis1-TMD are rejected before loading onto Get3; this explains the lack of Get3-dependence observed for these substrates in the overall targeting and insertion reactions in cell lysates.

Targeting of the Get3•TA complex to the ER and TA insertion into the membrane provides an additional selection filter (*Figure 9A*, Step 3). Our results show that two mechanisms can lead to further rejection of suboptimal substrates at this stage. TA proteins with suboptimal TMDs, such as 6AG, dissociate from Get3 more quickly than those with more hydrophobic TMDs; the rapid disassembly of the targeting complex could outcompete productive targeting and insertion processes. Importantly, this step provides the major mechanism to reject TA proteins with highly basic CTE, which are inserted into ER more slowly than those without (*Figure 9A*, step 3; and *Figure 9B–C*, blue). As the charged CTE compromised both Get3-dependent and Get3-independent TA insertions into ER, enrichment of basic residues in the CTE provides a general feature that enables TA proteins to escape insertion into the ER. In principle, transporting charges across a hydrophobic environment poses a high energetic barrier (*Engelman and Steitz, 1981*); this barrier could be imposed using either the phospholipid environment at the ER and/or translocases whose interior provide no compensation for these charges. In contrast to the ER, the mitochondrial membrane is enriched in anionic phospholipids such as cardiolipin (*Gebert et al., 2009*; *van Meer et al., 2008*) and might better tolerate a basic CTE on the TA protein. Finally, a four leucine substitution in the Fis1 TMD could compensate for its charged CTE and re-direct Fis1 to the GET pathway (*Wang et al., 2010*), suggesting that TA localization is specified by a combination of multiple physicochemical properties.

The GET pathway uses a variety of molecular strategies to select for correct substrates and reject suboptimal TAs. Selection during the initial substrate loading steps utilizes the difference in binding energy of various TAs for Sgt2 and for Get3. After TA proteins are loaded onto Get3, substrates can further partition between disassembly from Get3 and productive insertion into ER, during which authentic substrates partition more favorably into the productive insertion reaction compared to suboptimal substrates. As this partitioning occurs *after* irreversible ATP hydrolysis on Get3, it is analogous to proofreading mechanisms observed during rejection of near cognate tRNAs by the ribosome (*Rodnina and Wintermeyer, 2001a*, *2001b*). Although each mechanism provides a modest discrimination, the use of multiple, sequential selection steps allows the GET pathway to distinguish TA proteins destined to different organelles while accommodating a variety of amino acid sequences in the substrate protein. This principle has been observed with DNA and RNA polymerases (*Sydow and Cramer, 2009*), in translation elongation by the ribosome (*Ogle and Ramakrishnan, 2005*; *Rodnina and Wintermeyer, 2001b*), and in co-translational protein targeting mediated by the signal recognition particle (*Zhang et al., 2010*), and may represent a general principle by which key biological processes attain a high degree of fidelity.

Our results also suggest a role of Sgt2 in the selection of TA substrates into the GET pathway. Although the substrate preferences of Get3 parallels those of Sgt2, the differences in substrate

binding affinity of Get3 may not be fully realized in the GET pathway to generate effective selection. This is due to the high kinetic stability of the Get3•TA interaction, which dissociates more slowly ($\tau \sim$ 30–60 min) than its subsequent targeting and insertion ($\tau \sim$ 5–7 min). As a result, TA proteins that modestly destabilize the Get3•TA complex may not be effectively rejected based on differential binding (*Figure 9B*). Sgt2 could help overcome this problem by rejecting a fraction of GET-independent substrates prior to loading onto Get3. As Sgt2 forms less stable complexes with TA proteins than Get3, a borderline substrate bound to Sgt2 could more readily equilibrate with alternative machineries in the cytosol than if the same substrate were bound to Get3 (*Figure 9C*). This hypothesis, derived largely from energetic considerations, awaits to be tested rigorously in vivo.

Additional mechanisms could further enhance selection accuracy by the GET pathway. A simple extension of our model could include factors that compete with Get3 for receiving substrates from Sgt2, thus introducing a branch point upstream of Get3 that *irreversibly* directs suboptimal substrates from the GET pathway. The mammalian Sgt2 homolog, SGTA, associates with the Bag6 complex. Although the C-terminal domain of Bag6, together with TRC35 and Ubl4A, provides a structural analogue of the Get4/5 complex to mediate TA substrate transfer to TRC40 (the mammalian Get3 homologue) (*Mock et al., 2015*), Bag6 also contains additional domains that mediate membrane protein quality control (*Hessa et al., 2011*). Whether the Bag6 complex directs suboptimal TA substrates loaded on SGTA to quality control machineries and thus provides such a branch point is an attractive hypothesis that remains to be tested (*Lee and Ye, 2013*; *Leznicki and High, 2012*). Analogous branches have been suggested by physical interactions between Sgt2 and heatshock proteins in yeast (*Kohl et al., 2011*; *Wang et al., 2010*), but the mechanistic details of these and other machineries await discovery.

## Materials and methods

### Strains, plasmids, and transcripts

Yeast strains used for live-cell imaging are derivatives of W303 (ATCC201238) but were made *TRP1* and *ADE2* by repairing the endogenous auxotrophies. GFP-Fis1-tail constructs were made by PCR amplifying GFP(S65T) lacking a stop codon, with flanking SpeI and HindIII sites, the C-terminal 102 nt of Fis1 (plus additions for arginine codons and a stop codon included in the 3' oligo) with flanking HindIII and XhoI sites and cloning both fragments into p416ADH. mt-TagBFP is described in (*Okreglak and Walter, 2014*) and Sec63-tdTomato was a kind gift of Sebastian Schuck, ZMBH, Universität Heidelberg.

A model substrate is comprised of three tandem Strep tags at the N-terminus, a mutant yeast Smt3 in which the Ulp1 cleavage site was removed (a Pro insertion at residue 98 of Smt3), residues 207–222 of Bos1p (SEQTITSINKRVFKDK), various TMDs and CTEs defined in *Figure 2A*, and an opsin tag at the extreme C-terminus (GSMRMNGTEGPNMYMPMSNKTVD). TMD and CTE variants were constructed using QuikChange mutagenesis (Stratagene) or FastCloning (*Li et al., 2011*). For translation in yeast lysate, the coding sequences for model TA substrates were cloned into pSPBP6 (*Siegel and Walter, 1988*) under control of the SP6 promoter, and transcribed using a SP6 Megascript kit (Ambion). For coupled transcription-translation *in E. coli* lysate, the substrate coding sequences were cloned into the pCAT vector (*Kim and Swartz, 2001*) to replace that of chloramphenicol acetyl transferase.

Sfp-His$_6$ in a pET-29 vector was a gift from Jun Yin. For His$_6$-Sgt2, a His$_6$ tag and TEV protease cut site were fused to the N-terminus of full length Sgt2 and cloned into pET-33b. For untagged Sgt2, full length Sgt2 flanked by N-terminal and C-terminal TEV sites with a His$_6$-tag inserted downstream of the C-terminal TEV site was cloned into the pMAL-c2 vector as a C-terminal fusion to the malE gene. For ybbR-Get3, amplified DNA encoding wild-type *S. cerevisiae* Get3 was first cloned into a pET-28 vector as a C-terminal fusion to His$_6$-SUMO (a gift from André Hoelz) using SalI and NotI restriction sites. A ybbR tag (DSLEFIASKLA) was then inserted between residues S110 and D111 in Get3 through FastCloning (*Li et al., 2011*).

### Translation extracts

Yeast translation extracts and microsomes were prepared from △get3 or *SGT2FLAG*/△get3 (VDY 57; (*Wang et al., 2010*) strains as described in *Rome et al. (2013)*. *E. coli* S30 lysate was prepared and

coupled transcription-translation in the S30 extract was carried out as described (*Saraogi et al., 2011*), except that untagged T7 polymerase and untagged CmRS were used and anti-ssrA oligo was omitted. His$_6$-tagged T7 polymerase was used for the experiments in *Figure 8*.

## Fluorescence labeling

BODIPY-FL-CoA was synthesized and purified as described (*Yin et al., 2006*) with the exception that BODIPY-FL maleimide (Life Technologies) was used instead of Alexa Fluor 488 C$_5$ maleimide. The lyophilized compound was dissolved in DMSO, and dye concentration was quantified after dilution in methanol using $\varepsilon_{504} = 79,000$ M$^{-1}$cm$^{-1}$.

30 μM ybbR-Get3 was mixed with 60 μM BODIPY-FL-CoA and 12 μM Sfp-His$_6$ in Sfp labeling buffer (50 mM KHEPES, pH 7.4, 10 mM MgCl$_2$) in a total volume of 800 μL. The reaction mixture was rotated at room temperature for 1 hr. 10 μL 2 M imidazole (pH 7.0) was added before passing the reaction through Ni-NTA to remove Sfp-His$_6$. Gel filtration through a Sephadex G-25 (Sigma-Aldrich) column was used to remove excess BODIPY-FL-CoA and exchange ybbR-Get3$^{BDP}$ into GET buffer (50 mM KHEPES (pH 7.5), 150 mM KOAc, 5mM Mg(OAc)2, and 1 mM DTT). Translocation and ATPase reactions mediated by ybbR-Get3$^{BDP}$ were performed as described in *Rome et al. (2013)*.

## Protein expression and purification

Expression and purification of full length Get4/5 and His$_6$-tagged Get3 were performed as described (*Rome et al., 2013*). His$_6$-tagged Sfp was expressed and purified as described (*Yin et al., 2006*).

### Purification of His$_6$-Sgt2

Protein was expressed in BL21 Star (DE3) at 37°C for 4 hr after induction with 0.4 mM IPTG. Cell pellets were resuspended in lysis buffer (50 mM Tris-HCl (pH 7.5), 500 mM NaCl, 30 mM imidazole, and 5 mM β-mercaptoethanol (β-ME)) supplemented with 1X cOmplete Tablets EDTA free Protease Inhibitor Cocktail (Roche), Benzonase (Novagen), and lysed in 1X BugBuster (Novagen). Clarified lysates were loaded onto Ni Sepharose resin (GE Healthcare) and washed with 30 column volumes of lysis buffer. The protein was eluted using 50 mM Tris-HCl (pH 7.5), 500 mM NaCl, 300 mM imidazole, and then dialyzed against 50 mM KHEPES (pH 7.5), 150 mM NaCl, 20% glycerol for storage.

### Purification of untagged Sgt2

Protein was expressed and purified as with His$_6$-tagged Sgt2 with the following modifications: after affinity purification by Ni-NTA (Qiagen), protein was dialyzed against 20 mM Tris-HCl (pH 7.5), 20 mM NaCl, and 5 mM β-ME. TEV protease was included with partially purified Sgt2 to remove the MBP and His$_6$ tags. Samples were then incubated with amylose resin to remove MBP and uncut MBP-fusion proteins. The flowthrough was further purified by anion exchange MonoQ 10/100 GL (GE Healthcare) using a gradient of 20–550 mM NaCl, followed by gel filtration chromatography on a Superdex 200 16/60 (GE Healthcare) in GET Buffer.

### Purification of ybbR-Get3

Protein was expressed in *E. coli* BL21 Star (DE3) cells grown in LB media for 6–8 hr at 25°C after induction with 1 mM IPTG when cultures reached an A$_{600}$ ~ 0.3–0.6. The fusion protein was purified first using Ni-NTA affinity chromatography (Qiagen), and then His$_6$-tagged SUMO protease (gift from André Hoelz) was used to remove His$_6$-SUMO. The digestion mixture was passed through Ni-NTA to remove His$_6$-SUMO and SUMO protease. Dimeric Get3 was further isolated by gel filtration using a Superdex 200 16/60 (GE Healthcare). For ATPase assays, Get3 was further purified by anion exchange MonoQ 10/100 GL (GE Healthcare) before gel filtration chromatography.

### Purification of untagged T7 RNA polymerase

A PreScission protease cut site was introduced between the N-terminal His$_6$-tag and T7 polymerase. After purification of His$_6$-tagged T7 polymerase via Ni-NTA, the elution was dialyzed overnight against 50 mM Tris-HCl (pH 8.0), 200 mM NaCl, 5% Glycerol, and 10 mM imidazole at 4°C in the presence of His$_6$-tagged PreScission protease (a gift of André Hoelz). The mixture was passed through Ni-NTA to remove the PreScission protease and then further purified using a Superdex 200 16/60 (GE Healthcare). Purified T7 polymerase was stored in 50% glycerol at –30°C.

## Purification of untagged coumarin-tRNA synthetase (CmRS)

A PreScission protease cut site was introduced between the N-terminal $His_6$-tag and CmRS (*Charbon et al., 2011*; *Li et al., 2011*). After purification of $His_6$-tagged CmRS via Ni Sepharose (GE Healthcare), the elution was dialyzed overnight against 25 mM KHEPES (pH 7.5), 300 mM NaCl, 10% Glycerol, 10 mM $\beta$-ME, and 50 mM imidazole at 4°C in the presence of $His_6$-tagged PreScission protease (a gift of André Hoelz). The mixture was passed through Ni Sepharose to remove the PreScission protease and undigested CmRS. Purified CmRS was stored in 50% glycerol at –30°C.

## Purification of Sgt2•TA$^{Cm}$

7-hydroxycoumarin was incorporated into TA substrates using the amber suppression system described in *Charbon et al. (2011)*, *Saraogi et al. (2011)*. A TAG codon was introduced four residues upstream of the TMD. Coupled in vitro transcription-translation was carried out as described (*Saraogi et al., 2011*; Jewett and Swartz, 2004; Georke and Swartz, 2009) in the absence of anti-ssrA oligonucleotide and in the presence of 2 µM recombinantly purified Sgt2, untagged T7, and untagged CmRS. A 5 mL translation reaction was supplemented with 20 mM imidazole (pH 7.5), bound to 0.8 mL NiNTA agarose (Qiagen), and washed with 20 column volumes of GET Buffer containing 300 mM KOAc and 5 mM $\beta$-ME. Sgt2•TA complexes were eluted using GET buffer supplemented with 300 mM imidazole (pH 7.5) and 5 mM $\beta$-ME, concentrated (Amicon, 10K), and stored at –80 °C. For complexes containing Fis1-FisC and Fis1-BosC, excess Sgt2 was supplemented to a final concentration of 2 µM immediately after elution to maintain the TAs in a soluble complex with Sgt2. The presence of $His_6$-Sgt2 and 3xStrep-TA was verified by western blotting. The amount of Sgt2 in the purified complex was quantified by western blotting and standardizing against known amounts of purified $His_6$-Sgt2.

## Purification of Get3•TA complexes

For targeting and insertion reactions, 100 µL S30 translations were carried out at 30°C for TA substrates of interest in the presence of $^{35}$S-methionine and 2 µM untagged Sgt2. After translation, the reactions were supplemented with 2 mM ATP, 2 µM Get4/5, and 2 µM $His_6$-tagged Get3 to allow TA transfer to Get3 for 1 hr. Sample was diluted 1:1 with 2x capture buffer (1x capture buffer: 50 mM KHEPES, pH 7.5, 150 mM KOAc, 5 mM Mg(OAc)$_2$, 10% Glycerol, 5 mM $\beta$-ME, 20 mM Imidazole) and purified as for the Sgt2•TA complex. Elutions were concentrated to ~50 µL and TA concentrations were measured by scintillation counting. Samples were normalized to the same number of counts using GET buffer supplemented with 20 mg/mL BSA and 5 mM ATP.

Get3$^{BDP}$•TA$^{Cm}$ complexes were generated as above except that 5–10 mL translation-amber suppression reactions were carried out for TAG-containing TA constructs, and 500 nM Get3$^{BDP}$ was used. The mixture were loaded onto a 1 mL Strep-Tactin Sepharose (IBA Life Sciences) column and washed with 20 column volumes of GET buffer. Get3$^{BDP}$•TA complex was eluted using GET buffer supplemented with 2 mg/mL desthiobiotin. Elutions (10 mL) were concentrated through 10K concentrators (Amicon) and stored at –80°C. For 6AG, 500 nM Get3$^{BDP}$ was supplemented during washing and elution.

For the experiments in *Figure 8*, 100 µL S30 translations were carried out for TA substrates of interest as above, except $His_6$-tagged Sgt2, $His_6$-tagged Get4/5, and $His_6$-tagged Get3 were all added at the start of the reactions. The translation/transfer reactions were allowed to proceed for 2 hr, and then the reactions were loaded onto 50 µL Strep-Tactin Sepharose. The resin was first washed with 20 column volumes of GET buffer supplemented with 500 mM NaCl, and then washed with an additional 20 column volumes of GET buffer. The bound TA complexes were eluted with eight column volumes of GET buffer containing 100 mM KHEPES pH 7.5 supplemented with 30 mM biotin, concentrated to ~50 µL, and TA concentrations were measured by scintillation counting. Samples were normalized to the same number of counts using GET buffer without BSA or ATP.

### In vitro assays

### Overall targeting and translocation reactions in yeast lysate

Substrates of interest were translated for 1 hr in $\Delta get3$ lysate with or without recombinant Get3 present (as indicated in the text). Cyclohexamide and $\Delta get3$-derived microsomes were then added to initiate translocation. Substrates were allowed to translocate for an hour unless translocation time

courses were followed. Reactions were quenched by flash freezing in liquid nitrogen following by boiling in SDS buffer and analyzed by SDS-PAGE and autoradiography. The dependence of translocation efficiency on Get3 concentration for Bos1, 2AG, 4AG and Bos1-FisC were fit to **Equation 1**.

$$T_{obsd} = T_{max} \times \frac{[Get3]^h}{[Get3]^h + K_{1/2}^h}$$

(1)

in which $T_{obsd}$ is the observed translocation efficiency (%glycosylated TA) at a particular Get3 concentration, $T_{max}$ is the translocation efficiency at saturating Get3, $K_d$ is the Get3 concentration required for half maximal translocation, and $h$ is the hill coefficient. All curve-fitting were conducted using GraphPad Prism six for MacOS, GraphPad Software, San Diego California USA, www.graphpad.com.

## Substrate capture by Sgt2

100 μL S30 translations were carried out for each TA substrate of interest together with Ctrl (or Ctrl + 3xStrep) in the presence of $^{35}$S-methionine and 2 μM His$_6$-tagged Sgt2 at 30°C for 1 hr. Reactions were adjusted to 50 mM KHEPES, pH 7.5, 150 mM KOAc, 5 mM Mg(OAc)$_2$, 10% Glycerol, 5 mM $\beta$-ME, 20 mM Imidazole (capture buffer), and incubated (with rotation) with 50 μL Ni-NTA agarose equilibrated in capture buffer at 4°C for 1 hr. The mixture was loaded into a Mini Bio-Spin Column (Bio-Rad). The resin was washed with 2 mL of capture buffer and eluted with 300 μL capture buffer containing 300 mM imidazole. The load, flowthrough, and elution fractions were analyzed by SDS-PAGE and autoradiography. Images were quantified using ImageQuantTL (GE Healthcare). All capture efficiencies were normalized against that of the internal control (Ctrl or Ctrl + 3xStrep) translated and captured in the same reaction mixture.

## Equilibrium and kinetics of TA transfer

All proteins and complexes were ultracentrifuged (TLA100, Beckman Coulter Inc.) at 100,000 rpm for 30 min at 4°C prior to fluorescence measurements. Equilibrium titrations of TA transfer from Sgt2 to Get3 were carried out in GET buffer at 25°C in the presence of 20–50 nM Sgt2•TA$^{Cm}$ complex, 150 nM Sgt2, 150 nM Get4/5, 2 mM ATP, and indicated concentrations of Get3$^{BDP}$ in a Fluorolog-3–22 spectrofluorometer (Jobin Yvon). FRET efficiency (E) was calculated according to **Equation 2**,

$$E = 1 - \frac{F_{DA}}{F_D}$$

(2)

in which $F_{DA}$ is the fluorescence in the presence of donor and acceptor, and $F_D$ is the fluorescence of donor in the absence of acceptor.

The Get3 concentration dependence of the transfer reaction was fit to **Equation 3**,

$$E_{obsd} = E_{Max} \times \frac{[Get3]}{K_{1/2,app} + [Get3]}$$

(3)

in which $E_{obsd}$ is the observed FRET efficiency at a given Get3 concentration, $E_{Max}$ is the FRET efficiency at saturating Get3 concentrations, and $K_{1/2,app}$ is the concentration of Get3 required to reach half of the maximal FRET at the Sgt2 concentration used (150 nM).

Time courses of TA transfer from Sgt2 to Get3 were measured using a Kintek stopped-flow apparatus (Kintek Inc.). Reactions were initiated by mixing equal volumes of ~50 nM Sgt2•TA$^{Cm}$ in 150 nM Sgt2 with a solution containing 400 nM Get3$^{BDP}$, 400 nM Get4/5 and 2 mM ATP. For measurements in the presence of lysate, △get3 lysates were ultracentrifuged in a Beckman TLA 100.1 rotor for 1 hr at 100 k rpm, 4°C to remove ribosomes. TA transfer reactions were initiated by mixing equal volumes of ~50 nM purified Sgt2•TA$^{Cm}$ complex (supplemented with 100 nM Sgt2) with a mixture of 400 nM Get3$^{BDP}$, 400 nM Get4/5 and 2 mM ATP in ribosome-depleted lysate. Fluorescence decay of the FRET donor was monitored at 445 nm (445D40M band pass filter [Chroma]). Kinetic traces were biphasic and fit to **Equation 4**,

$$F_{obsd} = F_e + \Delta F_{fast} \times e^{-k_{fast}t} + \Delta F_{slow} \times e^{-k_{slow}t}$$

(4)

in which $F_e$ is the fluorescence when the reaction reaches equilibrium, $\Delta F_{fast}$ and $\Delta F_{slow}$ are the

amplitudes of the fluorescence changes in the fast and slow phases, respectively and $k_{fast}$ and $k_{slow}$ are the rate constants of the fast and slow phases, respectively. The overall halftime of the reaction, $t_{1/2}$, was obtained by noting the time at which the fluorescence change is 50% complete.

### Translocation of the Get3•TA complex

100 µL targeting and translocation reactions were initiated by adding 20 µL of $\Delta get3$ microsomes to the purified Get3•TA complex. At various time points, 10 µL samples were removed and quenched by addition of 2XSDS buffer and flash freezing in liquid nitrogen. Samples were analyzed by SDS-PAGE and autoradiography. The time course of translocation was fit to *Equation 5*,

$$\% translocation = T\left(1 - e^{-k_{obsd}t}\right) \tag{5}$$

in which $T$ is the %insertion at the end of the reaction, and $k_{obsd}$ is the observed rate constant of the targeting/insertion reaction.

In a given translocation reaction, the observed rate constant ($k_{obsd}$) and endpoint ($T$) are contributed by the rate constants of productive insertion ($k_{insert}$) and nonproductive reactions ($k_{nonproductive}$) according to *Equations 6–7*,

$$k_{obsd} = k_{insert} + k_{nonproductive} \tag{6}$$

$$T = \frac{k_{insert}}{k_{insert} + k_{nonproductive}} \times 100 \tag{7}$$

The values of $k_{insert}$ and $k_{nonproductive}$ were obtained by solving *Equations 6–7*.

### Kinetic Stability of the Get3•TA complex

Dissociation rate constants of Get3•TA complexes were measured by chasing 20–50 nM preformed $Get3^{BDP} \cdot TA^{Cm}$ complexes with a 10-fold excess of unlabeled Get3. The time course of loss of FRET was monitored and fit to *Equation 8*,

$$F = F_e + (F_0 - F_e)e^{-k_{dis}t} \tag{8}$$

in which $F$ is the observed donor fluorescence at a particular time, $F_0$ is the donor fluorescence at t = 0, $F_e$ is the donor fluorescence when the reaction is complete, and $k_{dis}$ is the dissociation rate constant for the Get3•TA complex.

## Cell imaging

Yeast strains were cultivated in SD –Trp lacking the appropriate nutrients for selection of episomal constructs at 25°C at early to mid-log phase ($OD_{600} \sim 0.3$–0.5). Cells were immobilized on coverslips coated with 0.1 mg/ml concanavalin A (Sigma) and imaged using a Nikon Eclipse Ti equipped with a spinning disk confocal (CSU-X1; Yokogawa), EMCCD camera (iXon3 897; Andor) and a 100×1.49 NA objective. Images were acquired with µManager software (*Edelstein et al., 2010*) and processed with ImageJ 1.49 (http://rsb.info.nih.gov/ij/).

## Acknowledgements

We thank Bil Clemons, Michael Rome, Kuang Shen, Xin Zhang, and members of the Shan and Clemons labs for critical discussions and comments on the manuscript, and the Dougherty lab for use of their HPLC. This work was supported by NIH grant GM107368 and Gordon and Betty Moore Foundation Grant GBMF2939 to SS, NIH grants R01GM32384 and U01GM098254 to PW, and the Leukemia and Lymphoma Society fellowship to VO. PW is an Investigator of the Howard Hughes Medical Institute.

# Additional information

## Funding

| Funder | Grant reference number | Author |
| --- | --- | --- |
| National Institutes of Health | GM107368 | Meera Rao<br>Un Seng Chio<br>Hyunju Cho<br>Shu-ou Shan |
| Howard Hughes Medical Institute | | Peter Walter |
| Gordon and Betty Moore Foundation | GBMF2939 | Shu-ou Shan |
| Leukemia and Lymphoma Society | | Voytek Okreglak |

The funders had no role in study design, data collection and interpretation, or the decision to submit the work for publication.

## Author contributions

MR, VO, Conception and design, Acquisition of data, Analysis and interpretation of data, Drafting or revising the article; USC, HC, Acquisition of data, Analysis and interpretation of data, Drafting or revising the article; PW, Conception and design, Drafting or revising the article; S-oS, Conception and design, Analysis and interpretation of data, Drafting or revising the article

## Author ORCIDs

Peter Walter, http://orcid.org/0000-0002-6849-708X
Shu-ou Shan, http://orcid.org/0000-0002-6526-1733

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
