## [Decision Letter]

Thank you for submitting your article "Multiple selection filters ensure accurate tail-anchored membrane protein targeting" for consideration by *eLife*. Your article has been reviewed by three peer reviewers, one of whom is a member of our Board of Reviewing Editors, and the evaluation has been overseen by Randy Schekman as the Senior Editor. The reviewers have opted to remain anonymous.

The reviewers have discussed the reviews with one another and the Reviewing Editor has drafted this decision to help you prepare a revised submission.

Summary:

The manuscript from Rao et al. describes a biochemical analysis of how the GET pathway distinguishes between closely related signals during targeting of tail-anchored membrane proteins to the ER and OMM. Previous studies implicate substrate transmembrane domain hydrophobicity and C-terminal basic charge as key features that allow discrimination. Here, the authors use an elegant in vitro approach to examine the role of distinct steps in the GET pathway, leading to a model where multiple steps contribute to fidelity. They also identify features of the targeting signals (hydrophobicity, helical content, C-terminal positive charge) that may be distinguished at different steps in the pathway.

Essential revisions:

1) The reviewers were convinced that there is clearly some discrimination of sub-optimal sequences during loading onto Sgt2. However, whether the key feature is helical content, hydrophobicity or some combination remains unclear (Figure 4). The authors should perform a simple computational analysis of ER and OMM TA protein TMs-according to their helical content model to determine whether calculated helical content is more predictive than hydrophobicity for the complete list of yeast TA proteins shown in Figure 1. If the author's model is valid they should see a correlation between lower helical content and OMM TA proteins compared to ER TA proteins.

2) The 'helical content' prediction (Figure 4) seems surprisingly low for these sequences given that they likely exist as helices in the membrane: only ~14% for Bos1, and less than 1% for most of the others. Are these surprisingly low helical content values explained by the choice of Agadir as the algorithm? The Agadir webserver lists three parameters (ionic strength, pH, temperature) that can be specified. The authors should indicate what parameters were used for their calculations and explicitly state that the calculated content is for a peptide in an aqueous environment. It seems odd that introducing a single glycine into 2AG reduces the helical content from 12% (2AG) to less than 1% (3AG). The reviewers also suggested that the authors test other algorithms that predict helical propensity.

3) All three reviewers had concerns with the conclusion that the Sgt2 to Get3 transfer step represents a second selectivity filter. The only significant effect is observed with Fis1-BosC and Fis1-FisC substrates, which raises two concerns: 1) these substrates were barely captured by Sgt2 in vitro (Figure 4). It wasn't clear how the complexes were even obtained for this experiment, or whether the complexes resembled those obtained with the optimal substrates. Were the TA segments of Fis1-BosC in a helical conformation and were they bound to the same surface of Sgt2? 2) More importantly, in yeast, Fis1 only binds Sgt2 indirectly, via heat shock proteins that bind to the Sgt2 TPR domain (Wang et al., 2010). Thus, in vivo, it never engages the C-terminal, met-rich substrate binding domain of Sgt2 and therefore never has a chance to transfer to Get3. The reviewers appreciate that the in vitro experiment described here lacks the yeast chaperones, but the physiologic significance of the result is unclear. It would be more compelling if a substrate that is efficiently captured by Sgt2 was transferred less efficiently to Get3. Instead, it seems that the vast majority of discrimination of Fis1-BosC and Fis1-FisC occurs upstream of the transfer reaction (i.e., at the initial Sgt2 loading step). For these reasons, the reviewers feel that this conclusion is not well supported, most likely not physiologically relevant, and should be de-emphasized in a revised version of this manuscript.

4) The Abstract and title of the manuscript would need to be modified to more accurately describe the findings. The role of the Sgt2-Get3 transfer step as a selectivity filet will need to be downplayed.

5) The third proposed selectivity step is proposed to occur during transfer of the substrate from Get3 to the membrane via Get1-Get2. Here, excess positive charge in the C-terminal region is proposed to slow the rate of membrane insertion (k-insert), thereby increasing the window of time for suboptimal substrates to dissociate from Get3 (k-nonproductive). This is an interesting proposal, but it was not clear whether the authors have considered other factors that might slow the insertion step. Suboptimal sequences may slow or prevent dissociation of Get3-substrate complexes from Get4/5 after transfer. The authors should test whether Get4/Get5 co-purify with Get3 TA protein complexes that contain optimal and non-optimal substrates. Could suboptimal sequences fail to activate the Get3 ATPase following dissociation of the complex from Get4/5? These possibilities would need to be checked before concluding that C-terminal charges slow the actual insertion step.

6) The authors should comment upon the finding that C-terminal charges reduce translocation for the Get3 independent substrates (Fis1-RR vs Fis1-RRRR). Could a reduced insertion rate for the highly charged tails be independent of the mechanism (uncatalyzed versus Get3/Get1/Get2 dependent) and be dependent upon ER lipid composition?

[Editors’ note: a previous version of this study was rejected after peer review, but the authors submitted for reconsideration. The first decision letter after peer review is shown below.]

Thank you for submitting your work entitled "Multiple selection filters ensure accurate tail-anchor membrane protein targeting" for consideration by *eLife*. Your article has been reviewed by three peer reviewers, one of whom is a member of our Board of Reviewing Editors, and the evaluation has been overseen by Randy Schekman as the Senior Editor. Our decision has been reached after consultation between the reviewers. Based on these discussions and the individual reviews below, we regret to inform you that your work was considered to be too preliminary to warrant publication in *eLife* at this point.

As you will see, the reviewers appreciated your efforts to biochemically dissect the TA substrate selection mechanism by the GET pathway. The identification of multiple 'checkpoints' along the post-translational GET pathway is certainly conceptually satisfying, and the use of innovative and quantitative assays appealing. However, all reviewers raised technical issues that require additional extensive experimentation. Among the main points are the suggestions to use a trap strategy to minimize direct loading onto Get3, to prove that the in vitro transfer reaction is really Get4/5-dependent, to establish that the Sgt2 material is of sufficient quality, and to more systematically analyze the behavior of a cleaner set of substrates. In addition, there are many other points that also need to be addressed. We would welcome a new submission on this topic when you are able to address the concerns of the reviewers and we would endeavor to approach the same experts to help evaluate such a new submission.

Reviewer #1:

This paper addresses the mechanism by which the hydrophobic tail of tail-anchored membrane (TA) proteins is recognized. This question is particularly relevant for the distinction between TA proteins targeted to the ER or mitochondria. Previous work has provided evidence that positive charged following the hydrophobic sequence are required for targeting to mitochondria, although the lack of such charges can be compensated for by changes in the hydrophobic region (Beilharz et al., 2003; Wang et al., 2010). In the present paper, the authors use different versions of the anchor sequence to investigate the effect of hydrophobicity and of the charged C-terminus on various steps of the GET pathway that is involved in TA protein integration into the ER. They conclude that Sgt2, the first component in the pathway, discriminates TAs on the basis of hydrophobicity. In addition, less hydrophobic TAs are only slowly transferred to Get3, the next component in the pathway. Finally, there is discrimination at the last step, the insertion of TAs into the membrane, in part because less hydrophobic TAs rapidly dissociate from Get3. At this point, TAs with positive charges also seem to be rejected.

The strength of the paper is the use of innovative and quantitative assays. Unfortunately, the results seem too preliminary to warrant several of the conclusions. Below is a list of the major concerns:

1) There is a general lack of systematic analysis. For example, the constructs used in Figure 2 for the in vivo analysis (beautiful data), are not employed for the in vitro analysis. Several of the constructs used in vitro have some problems or at least abnormalities. For example, Fis1-BosC is Get3-independent, and it is difficult to derive conclusions from it. The 4AG constructs is said to bind in a different conformation to Get3 based on a different steady state FRET level (Figure 7), so it may not be easily compared with other constructs. 6AG behaves very different from the other constructs in its dissociation rate from Get3 (Figure 9). Finally, only one construct with a change of the C-terminal charge is tested (Bos1-FisC). Why wasn't the C-terminus varied in the same way as in Figure 2? It is also surprising that no experiments were done with the actual mitochondrial protein Fis1. Because of the lack of systematic analysis, it remains questionable whether hydrophobicity and charges are the only variables that need to be considered.

2) Some of the results are not entirely novel. For example, the conclusion that Sgt2 recognizes hydrophobicity has been reached before.

3. The authors claim that C-terminal charges are recognized late in the Get pathway. However, an effect of the charges is also seen for the Get3-independent pathway, so how charges are recognized remains unclear. This means that the scheme in Figure 11 is at least incomplete. The paper does not really contribute to the major question in the field, i.e. how mitochondrial and ER TA proteins are discriminated.

Reviewer #2:

Molecular fidelity mechanisms that ensure accurate transmission of genetic information are well understood. By comparison, we know relatively little about mechanisms that make accurate protein sorting decisions on the basis of small differences in the biophysical properties of signal sequences. The basic steps by which TA proteins are guided to the ER membrane for insertion (GET pathway) but it remains poorly understood how they are sorted away from mitochondrial TA proteins. In yeast, Sgt2 and Get3 sequentially recognize ER signal sequences on TA proteins (partly in the TA transmembrane domain or TMD) but the raison d'etre for multi-step signal recogntion remains unclear. An appealing hypothesis is that Sgt2 can also transfer TA proteins for ubiquitination if they are not captured efficiently by Get3 and in doing so increase sorting fidelity. Intriguingly, Bag6 is a mammalian protein that scaffolds mammalian Sgt2 (SGTA), mammalian Get3 (TRC40, via interactions with additional pathway components), and a ubiquitin ligase that marks TA proteins for degradation in competition with TRC40-TA complex formation. By contrast, yeast lack an apparent Bag6 homolog or equivalent and there is a need for a better mechanistic framework describing how this organism achieves TA protein targeting fidelity.

In this paper, the authors seek to establish such a framework by measuring how the biophysical properties of tail-anchored (TA) proteins influence TA protein binding to Sgt2, TA protein transfer from Sgt2 to Get3, and the membrane-associated steps in the GET pathway. This paper's appeal is in the interesting questions it poses and in the development of fluorescence-based reporters to quantitatively measure TA transfer from Sgt2 to Get3 and TA protein dissociation from Get3. With the exceptions noted below, I find the in vitro analysis well performed and interpreted. However, the paper is thin on physiological relevance but that could be addressed by the additional experiments suggested below, which should raise the interest of this work for readers of *eLife*.

1) The authors show that suboptimal hydrophobicity of the TMD influences the kinetics (but not the equilibrium state) of TA transfer from Sgt2 to Get3 (Figure 4). This is an interesting observation but it warrants more supporting experiments because 4AG might be behaving differently from the other TAs in these in vitro assays for other reasons. Specifically, there are two independent concerns. First, the authors argue that "4AG was bound in a different conformation on Get3 compared to Bos1 and 2AG", and refer to their immunoprecipitation data in Figure S4 to argue that comparable amounts of Bos1, 2AG, and 4AG are transferred from Sgt2 to Get3. However, the blot images in Figure S4A seem to show that Get3 co-purifies significantly less 4AG than Bos1 or 2AG and are inconsistent with the quantitation in Figure S4B. More to the point, it is not shown that Sgt2 is even required for TA protein binding to Get3 under these in vitro conditions. Without repeating the experiment in the absence of Sgt2, the claim that these data support the conclusions about TA transfer from Sgt2 to Get3 based on the FRET assay is unsupported. One way to obtain corroborating evidence that TA transfer at equilibrium is the same for 4AG as Bos1 is to repeat this experiment with a photocrosslinking approach using a tRNA amber suppressor to introduce a photocrosslinker instead of a fluorophore and monitoring TA transfer from Sgt2 to Get3 (at different concentrations) by photocrosslinking. In fact, this approach could also be used to independently monitor the rate of TA transfer even on relatively short timescales because timepoints can be snap frozen and then crosslinked in the frozen state.

The second concern relates to the kinetic measurements of 4AG transfer. The authors don't measure the rate of spontaneous release of TA proteins from Sgt2. This rate, together with the rate of TA protein rebinding to Sgt2 complicates interpretation and might lead to in vitro effects that are not relevant in vivo because they are measured without competing cellular factors (e.g. factors for mitochondrial TA targeting). The authors should repeat the experiment shown in Figure 4 in the presence of the TA substrate "trap" used in the field (truncated Sgt2 containing the TA protein binding domain but lacking the Sgt2 dimerization/Get5-interaction domain). The TA trap should bind released substrates and preclude their re-binding to full-length Sgt2 for later handoff to Get3 (this is conceptually similar to the Get3-TA kinetic stability experiment in Figure 5). If the rate of 4AG dissociation from Sgt2 is much faster than Bos1, it would raise concerns about the in vivo relevance of these in vitro observations.

2) The authors argue that "Sgt2 plays a critical role….in enhancing discrimination against suboptimal substrates". It is difficult to accept this claim based on the data in Figure 6 because in the absence of Sgt2 the comparison with wild-type (in the absence of Get3 pre-incubation, which is the condition where 4AG is discriminated against) can't be made directly (ie. there is no detectable insertion). It is problematic to compare extracts prepared from wild-type cells with extracts prepared from deltaSGT2 cells and make mechanistic inferences on the basis of relatively small effects (less than 2-fold).

To remedy this issue, the authors should show that their purified Sgt2 rescues insertion to deltaSGT2 extracts and restores discrimination between Bos1 and 4AG. If the effect is indeed restored, the authors should exploit the add-back assay to test the role of Sgt2 interactions with Get5 by using Sgt2 point mutants that disrupt this interaction (Chartron et al. 2012).

Reviewer #3:

The manuscript by Rao et al., presents a biochemical dissection of the TA substrate selection mechanism by the GET pathway. This pathway must distinguish between closely related TA substrates for targeting and insertion into the ER or outer mitochondrial membranes (OMM). Previous work from a number of groups identified two critical substrate features that contribute to this discrimination-with TA substrates containing more hydrophobic TMDs tending to be ER-directed, and TA substrates containing less hydrophobic TMDs and sufficient positive charge at the extreme C-terminus tending to be OMM-directed. Using a largely in vitro approach the authors identify a series of steps in the GET pathway that they argue together contribute to fidelity in substrate selection:

1) As shown previously in Wang et al., 2010, the authors demonstrate that Sgt2 binds more tightly to TA proteins that have TMDs above a threshold level of hydrophobicity, than it does to those that do not. In addition, they demonstrate that the interaction with Sgt2 is insensitive to the presence (or absence) of significant C-terminal positive charge in the TA substrate.

2) Using a quantitative FRET assay to monitor the Sgt2-to-Get3 TA transfer reaction the authors argue for a second filter based on kinetic discrimination: more hydrophobic substrates are transferred to Get3 more quickly than less hydrophobic substrates.

3) Next, the authors present data suggesting that Get3 also loosely discriminates based on hydrophobicity (albeit less stringently than does Sgt2): below a threshold level of hydrophobicity, TA substrates more quickly dissociate from Get3, while above this threshold, they are more likely to remain bound.

4) Finally, the authors show that TA proteins containing sufficient positive charge at their C-terminus are less efficiently inserted into the bilayer.

The identification of multiple 'checkpoints' along the post-translational GET pathway is conceptually satisfying, and is similar to prior work in the co-translational system by the same lab. However, I have several concerns that should be addressed.

1) Recombinant Sgt2 is known to suffer from C-terminal degradation (e.g., Yeh et al., 2014). Because the C-terminal domain of Sgt2 contains the TA substrate binding region (Wang et al., 2010), degradation in this region may complicate the resulting kinetic analysis through, for example, formation of inactive (or less-active), truncated Sgt2 dimers that transfer substrates at different rates. The authors should show a coomassie stained gel to establish that their material is full-length. Note that Figure 5 shows a small portion of an Sgt2 western blot that already suggests the presence of degradation.

2) What is the basis of the Get3-independent insertion of the Fis1-BosC substrate shown in Figure 3?

3) In Figure 4 there appears to be a significant amount of TA substrate present in the Ni-NTA flow-through fraction. Is this free substrate or excess Sgt2-TA complex? Also, I am confused by the behavior of the Bos1 'Ctrl' substrate in Figure 4. If this is identical to the Bos1 substrate, except that it lacks the 3xStrep tag, shouldn't it bind to 6xHis-Sgt2 as efficiently as the 3xStrep-tagged Bos1 substrate?

4) Important controls are needed to demonstrate that the reconstituted Sgt2-to-Get3 substrate transfer reaction mimics the physiologic reaction-i.e., show Get4/5 dependence using appropriate point mutations, as has been done previously (Mock et al., 2015; Mateja et al., 2015). Direct TA loading onto Get3 can occur in vitro, and this might be exacerbated with the soluble SUMO substrates used here. Thus, it is important to distinguish between physiologic, Get4/5-dependent transfer from Sgt2 to Get3, and direct loading onto Get3, either from Sgt2 or from a pool of free, soluble TA substrate.

5) A related concern is that the fluorescent Sgt2-TA complexes, purified via a 3xStrep tag on the labeled substrate (Figure 5), are likely a mixture of Sgt2-TA and soluble TA substrate. Is this excess 'soluble' TA substrate competent for loading via Sgt2/Get4/Get5? Or via direct loading onto Get3? Are the kinetics of these steps identical to the loading kinetics of Get4/5-dependent transfer? As it stands, might this (and/or the concerns listed above) contribute to the biphasic transfer kinetics observed in Figure 7? If so, perhaps there is less substrate discrimination happening at the transfer step. A better way to do this is to purify the Sgt2-TA complex via an Sgt2 affinity tag rather than (or in addition to) the 3xStrep tag on the substrate (as was done in Figure 4)-this would help ensure that TA substrate is Sgt2 bound, which is likely a simpler starting point for characterization of the transfer reaction.

6) In Figure 8 and Figure 9 it appears they are monitoring TA binding to nucleotide-free Get3 (since bound nucleotide likely falls off during the Ni-pulldown (Figure 8) or the 'pulse-chase' experiment (Figure 9)). Under these apo conditions, 6AG is observed to dissociate quickly from Get3. But isn't it likely that the ADP bound state of the targeting complex is the physiologic one (as drawn in Figure 11)? Do the release kinetics for different substrates change if this experiment is done in the presence of MgADP? Perhaps 6AG release becomes slower, comparable to the other substrates? If so, this would indicate that Get3 plays even less of a role in substrate discrimination.

---

## [Author Response]

*Essential revisions:*

*1) The reviewers were convinced that there is clearly some discrimination of sub-optimal sequences during loading onto Sgt2. However, whether the key feature is helical content, hydrophobicity or some combination remains unclear (Figure 4). The authors should perform a simple computational analysis of ER and OMM TA protein TMs-according to their helical content model to determine whether calculated helical content is more predictive than hydrophobicity for the complete list of yeast TA proteins shown in Figure 1. If the author's model is valid they should see a correlation between lower helical content and OMM TA proteins compared to ER TA proteins.*

We carried out helical content analyses for ER and OMM TA proteins, and the Agadir scores for each TA are integrated in the revised Figure 1. As indicated by the Agadir scores, on average the TMDs of OMM TAs have lower helical propensity than ER-destined TAs. This information is described in the Results and Discussion.

It is not clear at this stage whether helical content or hydrophobicity scores are more predictive. When a wide range of substrates needs to be recognized with sufficient specificity, it is likely that the combination of multiple features contributes to recognition, and the contribution of each feature is dependent on the other feature. For this reason, we refrained from making more explicit statements on whether helical content is a more dominant determinant than hydrophobicity.

*2) The 'helical content' prediction (Figure 4) seems surprisingly low for these sequences given that they likely exist as helices in the membrane: only ~14% for Bos1, and less than 1% for most of the others. Are these surprisingly low helical content values explained by the choice of Agadir as the algorithm? The Agadir webserver lists three parameters (ionic strength, pH, temperature) that can be specified. The authors should indicate what parameters were used for their calculations and explicitly state that the calculated content is for a peptide in an aqueous environment. It seems odd that introducing a single glycine into 2AG reduces the helical content from 12% (2AG) to less than 1% (3AG). The reviewers also suggested that the authors test other algorithms that predict helical propensity.*

The Agadir algorithm calculates the helical content in aqueous solution. Model studies have shown that helical propensity generally increases in apolar environments, such as the membrane or when bound to protein, largely due to removal of competition from water for hydrogen bonding. However the difference in helical propensity between different TMDs still holds. The parameters for Agadir calculation are now included in the legend to Figure 1. We also added a note that these scores should be interpreted in a relative, rather than absolute sense (Figure 1 legend).

The disruption of helix formation from 2AG to 3AG results from the cooperative nature of helix formation. Consecutive i to i+4 hydrogen bonds reinforce one another to stabilize a helix, and at least two contiguous helical turns are needed to have a decent helical propensity (Creighton, Proteins, 2^nd^ ed., Chapter 5, pp 182-186). Insertion of the additional Gly in 3AG (in the middle of an IALILLII sequence) disrupted this minimal requirement.

We asked several experts about alternative algorithms for helical propensity calculation. The answer was that aside from running an MD simulation, Agadir is by far the most inclusive and accurate algorithm, as it also takes into account dipole effects, local side chain interactions that (de)stabilize the helix, etc. (Munoz and Serrano, 1994; Lacroix, et al., 1998). It has been tested on >1000 peptides against experimental data (in aqueous solution). As explained above, the reduction in helical content score from 2AG to 3AG is anticipated from the highly cooperative nature of helix formation and unlikely an artifact of the Agadir algorithm.

*3) All three reviewers had concerns with the conclusion that the Sgt2 to Get3 transfer step represents a second selectivity filter. The only significant effect is observed with Fis1-BosC and Fis1-FisC substrates, which raises two concerns: 1) these substrates were barely captured by Sgt2* in vitro *(Figure 4). It wasn't clear how the complexes were even obtained for this experiment, or whether the complexes resembled those obtained with the optimal substrates. Were the TA segments of Fis1-BosC in a helical conformation and were they bound to the same surface of Sgt2? 2) More importantly, in yeast, Fis1 only binds Sgt2 indirectly, via heat shock proteins that bind to the Sgt2 TPR domain (Wang et al., 2010). Thus,* in vivo*, it never engages the C-terminal, met-rich substrate binding domain of Sgt2 and therefore never has a chance to transfer to Get3. The reviewers appreciate that the* in vitro *experiment described here lacks the yeast chaperones, but the physiologic significance of the result is unclear. It would be more compelling if a substrate that is efficiently captured by Sgt2 was transferred less efficiently to Get3. Instead, it seems that the vast majority of discrimination of Fis1-BosC and Fis1-FisC occurs upstream of the transfer reaction (i.e., at the initial Sgt2 loading step). For these reasons, the reviewers feel that this conclusion is not well supported, most likely not physiologically relevant, and should be de-emphasized in a revised version of this manuscript.*

To answer the first question, Sgt2-TA complexes were purified via the tag on Sgt2, so the TAs eluted from the column are Sgt2-bound. For Fis1-FisC and Fis1-BosC, 2 µM Sgt2 were supplemented immediately after elution to maintain the TAs in complex with Sgt2 (described in the Materials and methods). We also ultracentrifuged the samples to remove any aggregates (including TAs that have dissociated from Sgt2) prior to all the fluorescence measurements. These procedures ensured that an Sgt2-TA complex was presented to Get3 during the measurement. Figure 6 shows that TA capture is completely lost with Sgt2△CTD, so these substrates are bound to Sgt2-CTD, but they are likely bound in a different conformation from that of a good TA substrate. Nevertheless, TA dissociation from Sgt2 is probable during the FRET measurement, in which the complex was diluted to an Sgt2 concentration of 150 nM (see more discussion in the next paragraph). Regarding the second question (Hsp70 vs Sgt2), we would like to point out that pull down experiments reflect the steady state distribution of the substrate on different factors in the cell or cell extract. By forcing Fis1 onto Sgt2 and watching its fate, our experiments attempt to understand, in part, the mechanism by which that steady state is achieved. Our results show that Fis1 bound Sgt2 weakly and even if it did, does not flux into the GET pathway; this is consistent with the steady-state observations of Fis1 accumulation on Hsp70 instead of Sgt2.

During the FRET measurement of TA transfer, the TAs partition between at least two fates: transfer to Get3 (monitored by FRET assay) and dissociation from Sgt2 (not directly observed). The absence of observed transfer with Fis1-FisC and Fis1-BosC indicate their highly unfavorable partitioning into the transfer reaction. This could arise from more rapid dissociation from Sgt2, less efficient transfer to Get3, or both. In the case that the absence of observed transfer arises exclusively from the lower kinetic stability of the Sgt2-TA complex, the mechanism becomes redundant with rejection at the Sgt2-TA binding step. We clarified these points in the Results and Discussion and toned down the conclusion that TA transfer could serve as a selection filter, as suggested by the reviewers.

*4) The Abstract and title of the manuscript would need to be modified to more accurately describe the findings. The role of the Sgt2-Get3 transfer step as a selectivity filet will need to be downplayed.*

We modified the Abstract and the model figure (new Figure 9) accordingly.

*5) The third proposed selectivity step is proposed to occur during transfer of the substrate from Get3 to the membrane via Get1-Get2. Here, excess positive charge in the C-terminal region is proposed to slow the rate of membrane insertion (k-insert), thereby increasing the window of time for suboptimal substrates to dissociate from Get3 (k-nonproductive). This is an interesting proposal, but it was not clear whether the authors have considered other factors that might slow the insertion step. Suboptimal sequences may slow or prevent dissociation of Get3-substrate complexes from Get4/5 after transfer. The authors should test whether Get4/Get5 co-purify with Get3 TA protein complexes that contain optimal and non-optimal substrates. Could suboptimal sequences fail to activate the Get3 ATPase following dissociation of the complex from Get4/5? These possibilities would need to be checked before concluding that C-terminal charges slow the actual insertion step.*

Since the Get3-TA complexes were previously purified using the tag on Get3, we agree that excess free Get3 could bring Get4/5 contaminations that introduce alternative possibilities. We think the most straightforward way to address these issues is to purify the Get3•TA complexes for Bos1, Bos1-FisC and Bos1-RRRR via the Strep_3_ tag on the TA substrate and re-perform the insertion assays in the absence and presence of ATP. As shown in the new Figure 8 (text on pages 15-16), the Get3•TA complexes purified via this procedure are free of Get4/5. Translocation assays using these complexes showed that slower insertions with Bos1-FisC and Bos1-RRRR are still observed in the absence of Get4/5, and the defects are similar with and without ATP present. These new data show that we can largely attribute the defect of the CTE mutants to targeting and insertion at the ER.

*6) The authors should comment upon the finding that C-terminal charges reduce translocation for the Get3 independent substrates (Fis1-RR vs Fis1-RRRR). Could a reduced insertion rate for the highly charged tails be independent of the mechanism (uncatalyzed versus Get3/Get1/Get2 dependent) and be dependent upon ER lipid composition?*

This is included in the Discussion as the following: “As the charged CTE compromised both Get3-dependent and Get3-independent TA insertions into ER, enrichment of basic residues in the CTE provides a general feature that enables TA proteins to escape insertion to the ER. In principle, transporting charges across a hydrophobic environment poses a high energetic barrier (Engelman and Steitz, 1981); this barrier could be imposed using either the phospholipid environment at ER and/or translocases whose interior provide no compensation for these charges.” We don’t currently know if the insertion of Fis1 CTE variants into ER is spontaneous or protein-dependent, and both lipid- and protein-dependent mechanisms could be envisioned with equal probability.

[Editors’ note: the author responses to the first round of peer review follow.]

[…]

*As you will see, the reviewers appreciated your efforts to biochemically dissect the TA substrate selection mechanism by the GET pathway. The identification of multiple 'checkpoints' along the post-translational GET pathway is certainly conceptually satisfying, and the use of innovative and quantitative assays appealing. However, all reviewers raised technical issues that require additional extensive experimentation. Among the main points are the suggestions to use a trap strategy to minimize direct loading onto Get3, to prove that the* in vitro *transfer reaction is really Get4/5-dependent, to establish that the Sgt2 material is of sufficient quality, and to more systematically analyze the behavior of a cleaner set of substrates. In addition, there are many other points that also need to be addressed. We would welcome a new submission on this topic when you are able to address the concerns of the reviewers and we would endeavor to approach the same experts to help evaluate such a new submission.*

[…]

Our original set of substrates (7) has been expanded to 13. We added two additional TMD variants, 3AG and 5AG, and four CTE mutants in which the number of charges is varied (Bos1-RR, Bos1-RRRR and Fis1-RR and Fis1-RRRR). Some of the major conclusions from the previous draft were substantiated by these additional substrate variants, that is, Sgt2 and Get3 are sensitive to variations in the TMD but not CTE, whereas the charged CTE is discriminated after substrate loading onto Get3 as the charges slow TA insertion into ER. In addition, by overcoming technical limitations in translation of the mitochondrial TA, Fis1p, we were able to further characterize this substrate in its transfer from Sgt2 to Get3 and show that Fis1 is completely rejected during this substrate handover. Finally, as natural amino acid substitutions tend to alter multiple physicochemical features, the larger set of TMD variants allowed us to perform an initial correlation analysis of capture efficiencies of Sgt2 with various features of the TMD. The result, surprisingly, indicated that hydrophobicity is not the sole determinant, and pointed at a significant role of the helical content of the TMDs in TA recognition by Sgt2. Collectivelly, the larger set of substrates corroborated some of the previous conclusions and gave unexpected new insights.

We provide two new pieces of data that address this concern. First, Figure 6 shows both the Get4/5-dependent and Get4/5-independent TA transfer reactions from Sgt2 to Get3; the latter is much slower than the former. As also shown in Figure 6 and Table 2, most of the TAs we tested (except for Fis1-TMD containing substrates) were transferred within 15 seconds and much faster than the Get4/5-independent reaction; thus these measurements reflect Get4/5-dependent TA transfer. Second, we re-performed the TA transfer reaction in the presence of Δget3 lysate, and found that transfer still occurred efficiently and rapidly (Figure 6). We did not use the suggested Sgt2ΔN ‘trap’ for two reasons: (i) there is no Sgt2ΔN in vivo, and we feel that the complete yeast lysate is a better mimic for TA transfer under physiological conditions; (ii) control experiments showed that Sgt2ΔN is not a passive trap for TA dissociation; instead, it significantly catalyzed TA release from Sgt2. The implication of this observation in terms of the mechanism of TA transfer is beyond the scope of this study, and we will pursue and publish it separately. But the bottom line is that we agree that re-examining TA transfer under more physiological conditions is a good concept, and did so using an alternative approach.

We have repurified Sgt2•TA complex using the affinity tag on Sgt2 instead of TA, as suggested by the reviewers. In addition, we have maintained a constant, excess amount of Sgt2 in all TA transfer assays ([TACm] ~ 50 nM and [Sgt2] = 150 nM). For TA substrates that are captured inefficiently by Sgt2, excess Sgt2 was also included during the washes and elutions during purification of the Sgt2•TA complex to maintain TA solubility prior to the assay. These optimizations in the procedure did improve the efficiency of the TA transfer reactions. As described above, we observed rapid TA transfer from Sgt2 to Get3 with favorable equilibrium (half-complete transfer at 5-10 nM Get3) and high FRET efficiency (~70-80%) for most of the substrates tested (Figure 6 and Table 2). In addition, gels for the purified Sgt2•TA complex are shown in Figure 5.

There are additional minor comments, but most of them are no longer applicable with the new sets of data and with changes in some of the conclusions. The only significant comment I would address upfront is Reviewer 2’s suggestion to use an alternative, crosslinking approach instead of FRET to measure TA transfer from Sgt2 to Get3. This is conceptually the same experiment and, as shown in Figure 6, most of the Get4/5-dependent transfer reactions are rapid and complete within the timescale of reliable manual mixing (τ ~5-15 sec); so it is necessary to use a continuous fluorescence assay coupled to rapid mixing devices to accurately make these measurements. Finally, the Get4/5-independent TA transfer reaction corroborated that the observed FRET is due to TA loading on Get3, rather than the TA being close to Get3 in a TASgt2-Get4/5-Get3 transfer complex.